DOI: 10.1038/ncomms15650 · OPEN

# Simple and scalable growth of AgCl nanorods by plasma-assisted strain relaxation on flexible polymer substrates

Jae Yong Park[1], Illhwan Lee[1], Juyoung Ham[1], Seungo Gim[1] & Jong-Lam Lee[1]

Implementing nanostructures on plastic film is indispensable for highly efficient flexible optoelectronic devices. However, due to the thermal and chemical fragility of plastic, nanostructuring approaches are limited to indirect transfer with low throughput. Here, we fabricate single-crystal AgCl nanorods by using a $Cl_2$ plasma on Ag-coated polyimide. Cl radicals react with Ag to form AgCl nanorods. The AgCl is subjected to compressive strain at its interface with the Ag film because of the larger lattice constant of AgCl compared to Ag. To minimize strain energy, the AgCl nanorods grow in the [200] direction. The epitaxial relationship between AgCl (200) and Ag (111) induces a strain, which leads to a strain gradient at the periphery of AgCl nanorods. The gradient causes a strain-induced diffusion of Ag atoms to accelerate the nanorod growth. Nanorods grown for 45 s exhibit superior haze up to 100% and luminance of optical device increased by up to 33%.

[1] Department of Materials Science and Engineering, Pohang University of Science and Technology (POSTECH), Pohang 790-784, Korea. Correspondence and requests for materials should be addressed to J.-L.L. (email: jllee@postech.ac.kr).

Flexible plastic substrates are lightweight, inexpensive and enable roll-to-roll (R2R) mass production[1–3], so they have applications as components in next-generation optoelectronic devices such as organic light-emitting diodes (OLEDs)[1,3–5], displays[6,7], organic solar cells[8–11] and photodiodes[2,12]. To improve the efficiency of devices based on flexible substrates, nanostructuring technology has become indispensable, because planar structure causes unwanted surface reflection (about 10.4% for polymer film with refractive index $n = 1.6$) and total internal reflection (89% for critical angle 38.7°). Non-planar nanostructure has optical properties that control its light propagation, reflection, diffraction and absorption features[4,5,13–15].

Efforts to fabricate nanostructures on rigid substrates include top-down methods such as photo-[16,17], laser interference-[18] and e-beam lithography[19]; and bottom-up methods such as synthesis of semiconductor nanorods (NRs) by vapour–liquid–solid (VLS)[20–22], chemical vapour deposition (CVD)[23,24] and thermal oxidation[25]. However, most of these method are not suitable for fabricating nanostructure on flexible substrates. Lithography techniques use photoresists; their removal entails use of solvents that can damage a polymer film[26,27]. VLS, CVD and thermal oxidation require growth temperatures $> 500\,^{\circ}C$[28–30], but most plastic substrates cannot tolerate temperature $> 150\,^{\circ}C$[31].

To realize fabrication of nanostructures on flexible substrates, it will require simple fabrication methods that can be conducted at low temperature. Due to the chemical and thermal fragility of the polymer, indirect nanostructure-transfer methods such as nanoimprint lithography (NIL)[32–34], and Langmiur–Blodgett methods[35,36] are widely used to fabricate nanostructures on plastic substrates. Recently, NIL has been used to form sub-wavelength structure with period of 250 nm and depth of 50 nm on polyethylene terephthalate (PET) films; because this structure minimized the waveguide mode in the substrate, OLEDs had 84% higher power efficiency than a device fabricated on planar film[37]. However, NIL has limited applicability to ultraviolet or thermal resins such as polymethyl methacrylate and polydimethylsiloxane[32–34]. Also, the method has low throughput, is complex and expensive, and uses master moulds that may deform after repeated use. The Langmiur–Blodgett process can transfer nanostructures with targeted density and thickness by immersing the substrates in a solution, but requires nanoparticles that have been functionalized with surfactant, and a subsequent careful dipping–pulling process. In addition, NRs fabricated by this method can be aligned only in lateral directions[38], so the spatial density of NRs on the surface is low. Such problems could be solved by developing a plasma-assisted method to induce strain relaxation, which is the driving force of NR growth on polymer substrate. A polymer substrate flexes convexly during the growth of NRs, so migration of Ag atoms is accelerated by strain-induced diffusion; as a result, NRs grow rapidly on the polymer substrate.

Here, we demonstrate the growth of single-crystalline AgCl NRs by using a $Cl_2$ plasma source. Vertically well-aligned AgCl NRs were grown by a strain-relaxation process on several kinds of transparent substrate. To understand the growth procedures of AgCl NRs on Ag film, synchrotron radiation X-ray photoelectron spectroscopy (XPS) and X-ray diffraction (XRD) data were systematically examined with respect to plasma exposure time and various kinds of substrates. Spontaneous flexion of substrates during growth increases both the supply of Cl radicals and diffusion of Ag. From the off-axis phi-scan analysis and high-resolution transmission electron microscopy (HR-TEM) images, we propose a detailed growth structure, that is, AgCl NRs (200) grow epitaxially in cube-on-hexagon geometry on Ag (111) to minimize the strain energy.

## Results

**Fabrication of AgCl NRs on various substrates.** Single-crystalline AgCl NRs were grown on the transparent substrate with deposited Ag layer by introducing a chlorine plasma source. We deposited 300-nm-thick Ag on 100-μm-thick colourless polyimide (PI) film, then exposed the Ag to $Cl_2$ plasma (Fig. 1). During the plasma treatment, Ag atoms reacted with highly reactive Cl radicals on the surface of Ag film; as a result AgCl formed, as confirmed by XRD (Supplementary Fig. 1).

The plasma condition has an important influence on the morphology of the AgCl nanostructures, so we optimized the plasma power ($P$) and process pressure ($p$) to produce well-defined AgCl NRs (Supplementary Figs 2 and 3). At $P = 50\,W$, surface morphology of Ag was almost unchanged, because Cl radicals did not have sufficient energy. At $P = 200\,W$, the surface began to develop nanodots; at $P = 350\,W$, AgCl NRs were grown successfully, but at $P > 350\,W$, they began to shrink and aggregate. We then optimized $p$ while maintaining $P = 350\,W$ (Supplementary Fig. 3). At $p = 2\,mTorr$, NRs produced were short due to insufficient supply of Cl radicals. At $p = 10\,mTorr$, sufficient amount of Cl radicals led to longer NRs. However, at $p = 10\,mTorr$ the NRs began to aggregate because of excessive supply of Cl radicals. Therefore, we identified $P = 350\,W$ and $p = 10\,mTorr$ for optimal growth of AgCl NRs.

Plasma exposure time can also affect the morphology of AgCl NRs (Fig. 2a). When Ag on PI film was exposed for 15 s, numerous AgCl nanodots were produced. At exposure time of 30 s, the nanodots began to grow to form NRs with length of 1 μm. Then, their length gradually increased until 60 s. As the exposure time increased over 90 s, the NRs became severely deformed to hierarchical structures. This simple method of NR growth could be applicable to any substrates including PET (film thickness = 130 μm) (Fig. 2b) and soda-lime glass (700 μm) (Fig. 2c), but the growth rates strongly depends on the type of substrate. The growth behaviour of AgCl NR on PET film was similar to that on PI film, but the NRs grew slowly on 700-μm-thick glass (Fig. 2c).

**Optical properties of AgCl NRs.** The size and optical properties of the NRs can be controlled by changing Ag thickness ($t_{Ag}$) (Fig. 3a). At $t_{Ag} = 10\,nm$, AgCl nanostructures were parabolic. As $t_{Ag}$ increased, AgCl NRs became aligned orthogonal to the substrate. The surface morphology was examined with $t_{Ag}$ and plasma exposure time and summarized in Supplementary Fig. 4. The distributions of period and diameter were plotted and an

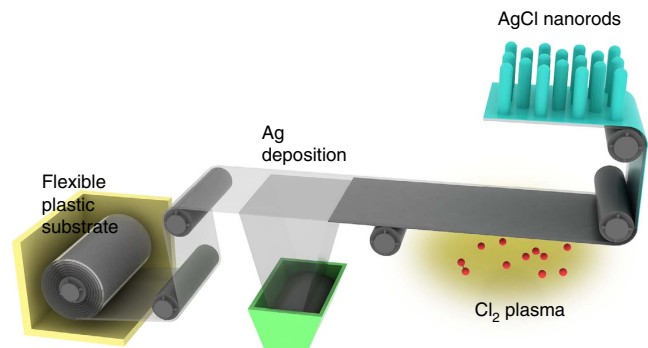

**Figure 1 | Schematic diagram of fabrication of AgCl nanorods on flexible plastic substrates.** Ag was deposited on flexible plastic substrates. Ag-coated substrates were exposed to $Cl_2$ plasma. During plasma treatment, Ag was reacted completely to produce AgCl nanorods on flexible plastic substrates. This growth method is compatible with roll-to-roll processing.

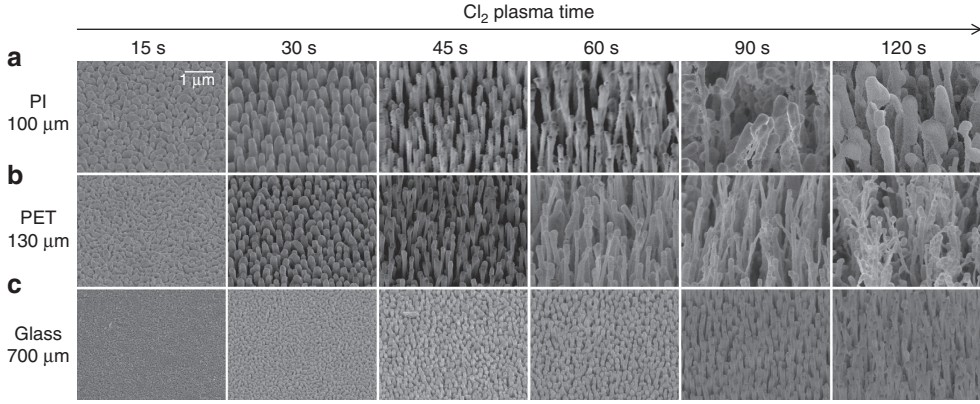

**Figure 2 | AgCl nanorods on different substrates.** Time evolution of Ag film during $Cl_2$ plasma treatment on different thick substrates, (**a**) 100-μm-thick polyimide (PI), (**b**) 130-μm-thick polyethylene terephthalate (PET) and (**c**) 700-μm-thick soda lime glass. Scale bar, 1 μm.

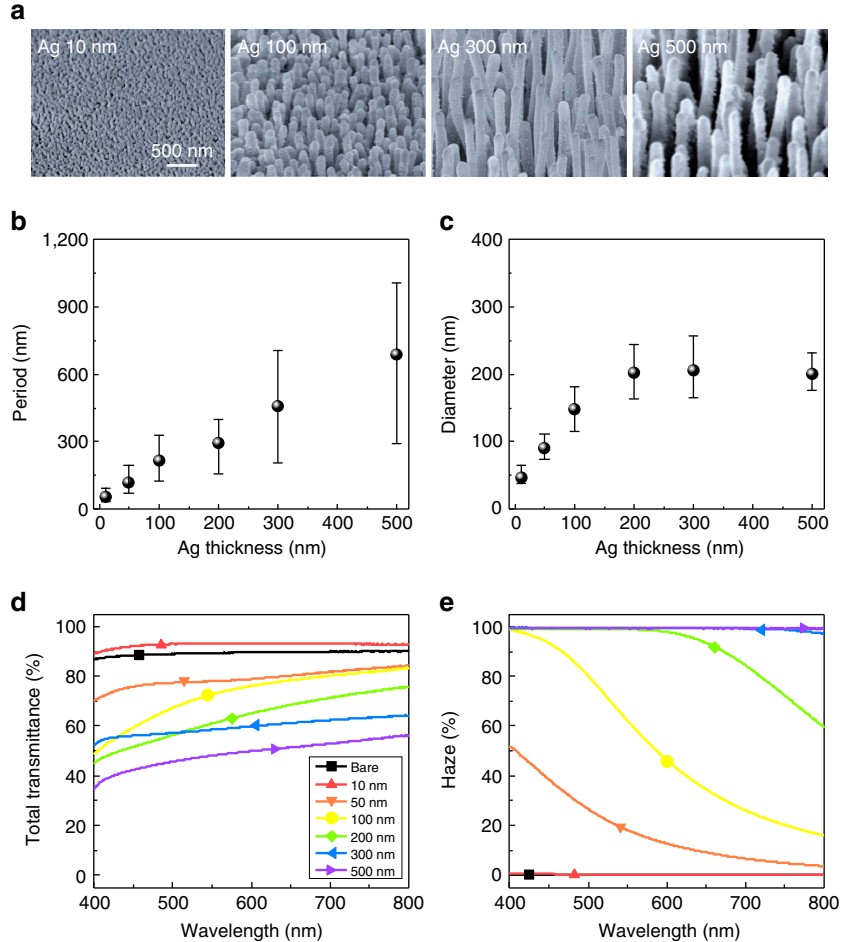

**Figure 3 | Surface morphology of AgCl nanorods and its optical properties.** (**a**) Scanning electron microscopy (SEM, 45° tilt view) images of AgCl nanostructures with sizes of 10, 100, 300 and 500 nm Ag (from left to right). Plasma treatment was fixed at 45 s. (**b**) Average periods and (**c**) average diameters of AgCl nanostructures as a function of Ag thickness. (**d**) Total transmittance and (**e**) haze transmittance of bare polyimide film and various AgCl nanostructures.

average period and the average diameters were determined (Supplementary Fig. 5). As $t_{Ag}$ increased from 10 to 500 nm, the average period changed linearly from 60 to 690 nm (Fig. 3b). The average diameter increased until $t_{Ag} = 200$ nm, then saturated (Fig. 3c).

The optical properties of size-tunable AgCl NRs were examined by using an integrating sphere to measure total transmittance ($T_{total}$) and Haze ($H$) (Fig. 3d,e). Over the wavelength range $400 \leq \lambda \leq 800$ nm, the as-received PI film had average $T_{total} = 89.4\%$ and $H = 0.22\%$. The sub-wavelength scale AgCl ($t_{Ag} = 10$ nm) had the highest average $T_{total} = 93.2\%$ and average $H = 0.27\%$, by providing gradually changed refractive index and allowing only zeroth-order transmitted light[39,40]. In contrast, the $T_{total}$ of the other AgCl structures decreased and $H$ increased as $t_{Ag}$ increased. $H$ was $\sim 100\%$ at $t_{Ag}$ $> 300$ nm.

The wavelength-scale structures induced high-order diffraction modes of transmitted waves, and thereby increased $H$. These AgCl NRs are suitable for substrates in OLED-illumination applications because high $H$ causes scattering, which improves the outcoupling efficiency of confined wave-guided light in the substrates. $T_{total}$ and $H$ were also measured as functions of $t_{Ag}$ and $Cl_2$ plasma time (Supplementary Fig. 4b,c). Consequently, the light-scattering characteristics can be managed from haze-free to fully scattering by controlling the size of the AgCl nanostructures, which can be tuned simply by adjusting $t_{Ag}$.

**Surface analysis of AgCl NRs.** The Ag $3d$ spectrum of as-deposited Ag film showed peaks centred at 374.0 eV and 368.0 eV (Fig. 4a), which are ascribed to Ag $3d_{5/2}$ and Ag $3d_{3/2}$, respectively. These two peaks could each be separated into two groups. The peaks at 374.0 and 368.0 eV could be assigned to metallic $Ag^0$ (refs 41,42), and the peaks at 373.6 and 367.6 eV could be attributed to $Ag^+$ of native silver oxide[43,44]. After Ag was exposed to the $Cl_2$ plasma for 45 s, peaks centred at 373.7 and 367.7 eV appeared; these can be ascribed to $Ag^+$ of the AgCl (refs 45,46). The Cl $2p$ spectrum (Fig. 4b) showed no signal before the plasma exposure, but afterward the spectrum showed peaks centred at 199.4 and 197.8 eV, which are ascribed to Cl $2p_{3/2}$ and Cl $2p_{1/2}$, respectively[47].

To investigate electrical properties during the transition from Ag to AgCl, we measured current–voltage ($I$–$V$) curves of plasma-exposed Ag film on sapphire as a function of the $Cl_2$ plasma exposure time (Fig. 4c). In all cases, $I$ increased linearly with $V$; in as-deposited Ag film, the resistance (inverse of slope) was 1.97 Ω. The resistance gradually increased with the plasma exposure time and reached $8.32 \times 10^{10}$ Ω at 90 s. This trend is evidence that $Cl_2$ plasma treatment converts the Ag conducting layer to an AgCl insulating layer. These results suggest that treatment for 90 s caused the Ag to react completely to produce AgCl.

**Crystallography of the AgCl NRs.** The crystal structures of the AgCl NRs (Fig. 5a) were characterized using transmission electron microscopy (TEM). Selected area electron diffraction (SAED) patterns (inset) showed a few defects; they were caused by decomposition of AgCl to polycrystalline Ag under the high-energy electron beam[48,49]. HR-TEM images of the AgCl NRs show well-aligned rock salt single crystals (Fig. 5b,c). The NR grew along the [200] direction. The measured interplanar distances were 2.77 and 3.20 Å, which are consistent with those of (111) and (200) AgCl planes, respectively.

The growth direction of the NRs was also confirmed by XRD; the patterns (Fig. 5d) of as-deposited Ag on the PI film and $Cl_2$ plasma-treated Ag were measured in the Bragg–Brentano configuration ($\theta$–$2\theta$ geometry). In as-deposited Ag film, only Ag peaks exist and the peak of (111) crystallographic plane was dominant. After plasma treatment, only AgCl peaks existed, and diffraction on the (200) planes reached its maximum intensity. The Bragg–Brentano scan represents only crystallographic planes of the film that are parallel to the substrate, so these results indicate that large amounts of (200) planes of AgCl were parallel to the PI film. Because the NRs were aligned vertically on the substrate, their growth direction of the NRs was determined to be perpendicular to the (200) planes.

This preferential growth could have two explanations. First, movement of Cl atoms into Ag may be fastest in the [200] direction because AgCl (200) has the maximum interplanar distance. Second, the AgCl NRs grow along the [200] direction to minimize strain energy because the lattice mismatch $(d_{AgCl} - d_{Ag})/d_{Ag}$ is the smallest (10.8%) between Ag (111) and AgCl (200) along Ag[−211]||AgCl[010]. The other diffraction

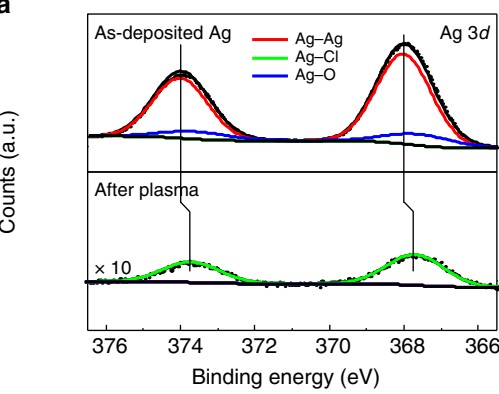

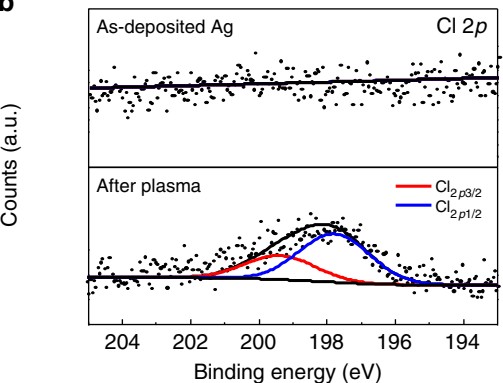

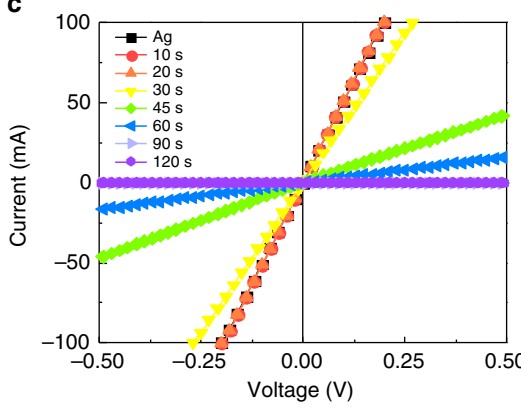

**Figure 4 | Characterization of AgCl nanorods.** X-ray photoelectron spectroscopy (XPS) of (**a**) Ag $3d$ and (**b**) Cl $2p$ core spectra for as-deposited Ag (top) and $Cl_2$ plasma exposed Ag (bottom). (**c**) Current–voltage characteristics of AgCl nanostructures as a function of $Cl_2$ plasma exposure time.

peaks in XRD may originate from the thin layer that connected the NRs.

To further confirm the epitaxial relationship between Ag and AgCl, off-axis phi scans (azimuthal scan) were performed. XRD phi scan was used to determine the crystal orientations of Ag and AgCl. To measure off-axis phi scans, we choose an arbitrary reflection plane which is not surface normal plane. To allow interpretation of the in-plane relationship, c-Sapphire was used instead of PI film. The surface normal planes of c-sapphire, Ag and AgCl are (006), (111) and (200) respectively. Thus, we performed phi scans along the c-sapphire (113) plane with psi angle ($\psi$) = 61.2°, Ag (311) with $\psi$ = 29.5° and AgCl (220) with $\psi$ = 45°, where $\psi$ is the angle between measured reflection plane

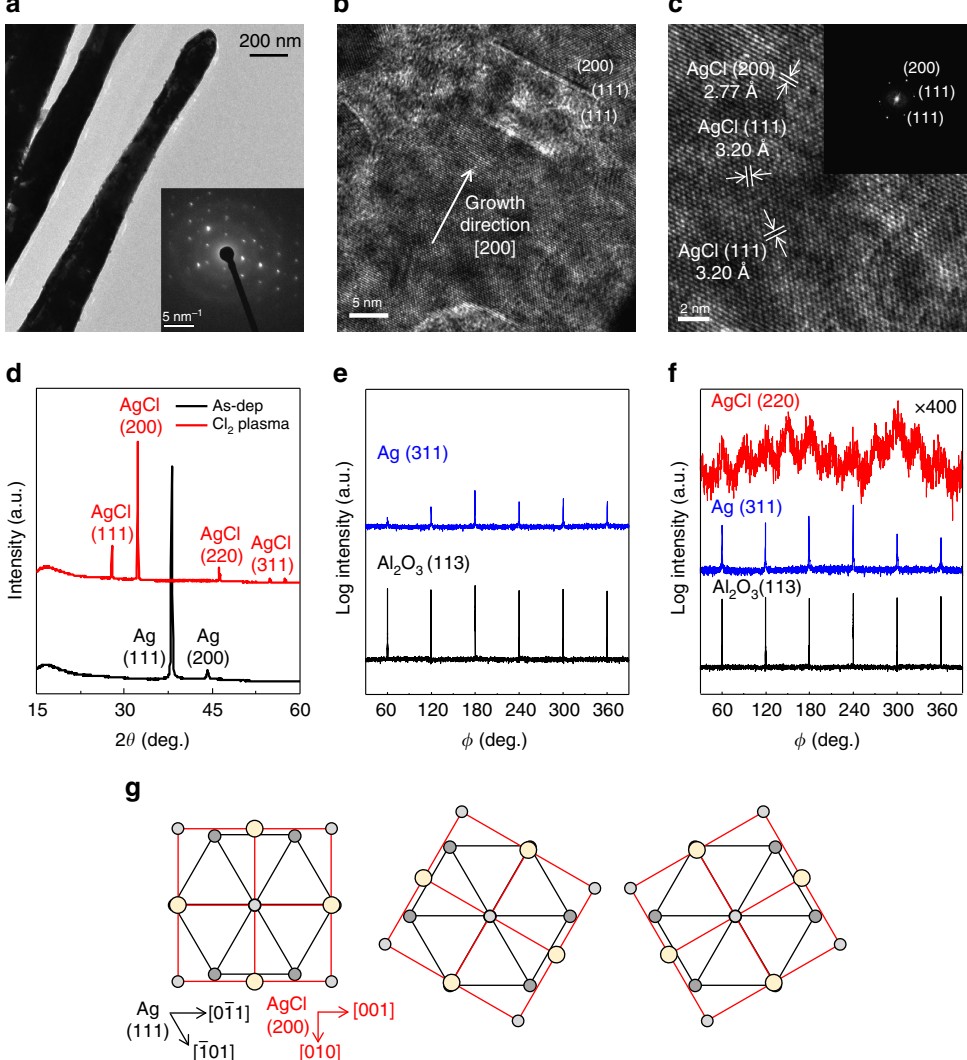

**Figure 5 | Crystallography of the AgCl nanorods.** (**a**) transmission electron microscopy (TEM) images of single AgCl nanorod and selected area electron diffraction (SAED) patterns (inset). SAED pattern recorded along [011] zone axis. (**b,c**) High-resolution TEM (HR-TEM) image of AgCl nanorod. Nanorod grows along [200] direction. (111) and (200) planes of AgCl are noted. (**d**) X-ray diffraction (XRD) patterns of as-deposited Ag (black) and Cl$_2$-plasma-exposed Ag (red). Azimuthal diffraction scans of 300-nm of Ag on c-sapphire for AgCl (200), Ag (311) and Al$_2$O$_3$ (113) reflection (**e**) as-deposited and (**f**) after Cl$_2$ plasma exposure. (**g**) Atomic configuration in the Ag/AgCl system with Ag (111)//AgCl (200) planes with 60° azimuthal rotation of AgCl (200).

of the phi scan and the surface normal plane of sample. In the phi scan, Ag film exhibited six-fold symmetry on c-sapphire substrates with coincidence in angular positions (Fig. 5e). This result reveals that Ag film was deposited on c-sapphire with (111)-preferred orientation. After Cl$_2$ plasma treatment, Ag changed to AgCl. Ag maintains the six-fold symmetry, but AgCl exhibits 12-fold symmetry with 30° intervals (Fig. 5f). Ag and AgCl had the same angular positions. AgCl (200) with 12-fold symmetry has three types of cubic domains, which are aligned on hexagonal Ag (111) planes with 60° azimuthal rotation (Fig. 5g). The phi scan confirmed that the growth direction of AgCl NRs was (200), which is in good agreement with HR-TEM observation (Fig. 5b).

**Driving force of growth of AgCl NRs.** On the basis of these results, we proposed a growth mechanism of the AgCl NRs (Fig. 6). When Ag was deposited, the face-centred-cubic Ag film tended to grow with the preferred orientation of (111) (Fig. 5d). During the initial stage of growth, Ag atoms on the surface of Ag

film reacted with Cl radicals from the Cl$_2$ plasma (Fig. 6a). Conversion of Ag to AgCl nanodots imposes a large strain because of the lattice mismatch between Ag (lattice constant $a_{Ag} = 4.079$ Å[3]) and AgCl ($a_{AgCl} = 5.545$ Å) (Fig. 6b). To minimize the strain energy, the nucleated AgCl nanodots had (200) orientation, which has the smallest lattice mismatch with Ag (111) film. Because the AgCl nanodots acted as a seed for the growth of AgCl NRs, the Ag atoms reacted with adsorbed Cl radicals at the interface of the NRs with Ag film (Fig. 6c). At that time, the AgCl underwent a compressive strain from the remaining Ag (Fig. 6e); to relieve this strain, NRs grew vertically. As a result, the basal plane of the NR was subjected to less strain than the surface area of AgCl between the NRs, so a strain gradient developed in the AgCl layer. This strain gradient induced strain-induced diffusion of Ag atoms in Ag film to the basal plane of the NR (Fig. 6f); this migration allowed selective growth of NRs at the bottom of the nucleated AgCl nanodot. In this mechanism, the supply of Cl radicals is constant, independent of the kind of substrate. Thus, diffusion of Ag atoms to the base of NRs determines their growth rates.

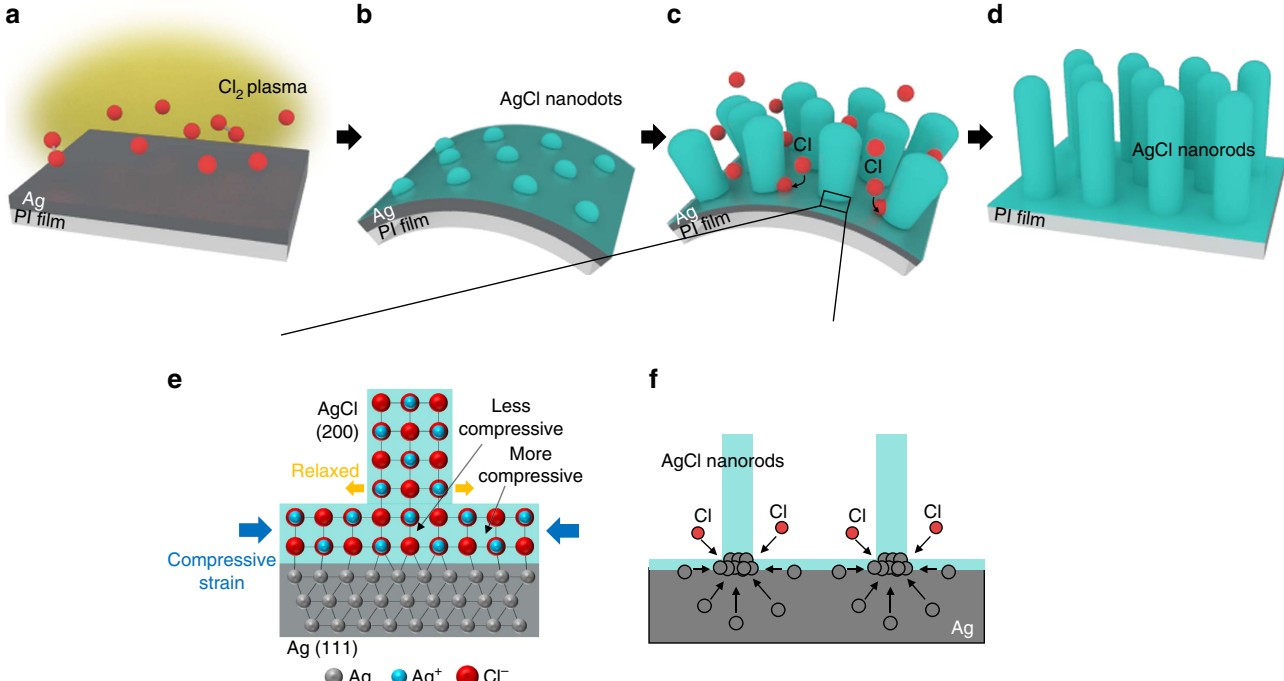

**Figure 6 | Schematic illustration of strain-relaxation driven growth mechanism of AgCl nanorods.** (**a**) Ag-coated PI film was exposed to $Cl_2$ plasma. (**b**) In early stage of growth, Ag surface reacted with Cl radicals, producing AgCl islands due to volume expansion from Ag to AgCl. PI film bent convexly in response to strain. (**c**) Cl radicals diffused into Ag/AgCl interfaces; as a result, compressive strain was applied to AgCl and acted as driving force of nanorod growth. Interfacial lattice mismatch (Ag/AgCl) guided (200)-preferred grow of AgCl nanorod. (**d**) AgCl nanorods grow on PI film as a result of strain relaxation. (**e**) Atomic configuration of Ag/AgCl interface. Compressive strain in the AgCl stimulates growth of nanorods. Base of nanorod was subjected to less strain due to strain relaxation of nanorods. (**f**) During growth of nanorod, Ag atoms diffuse to its base because of strain gradient.

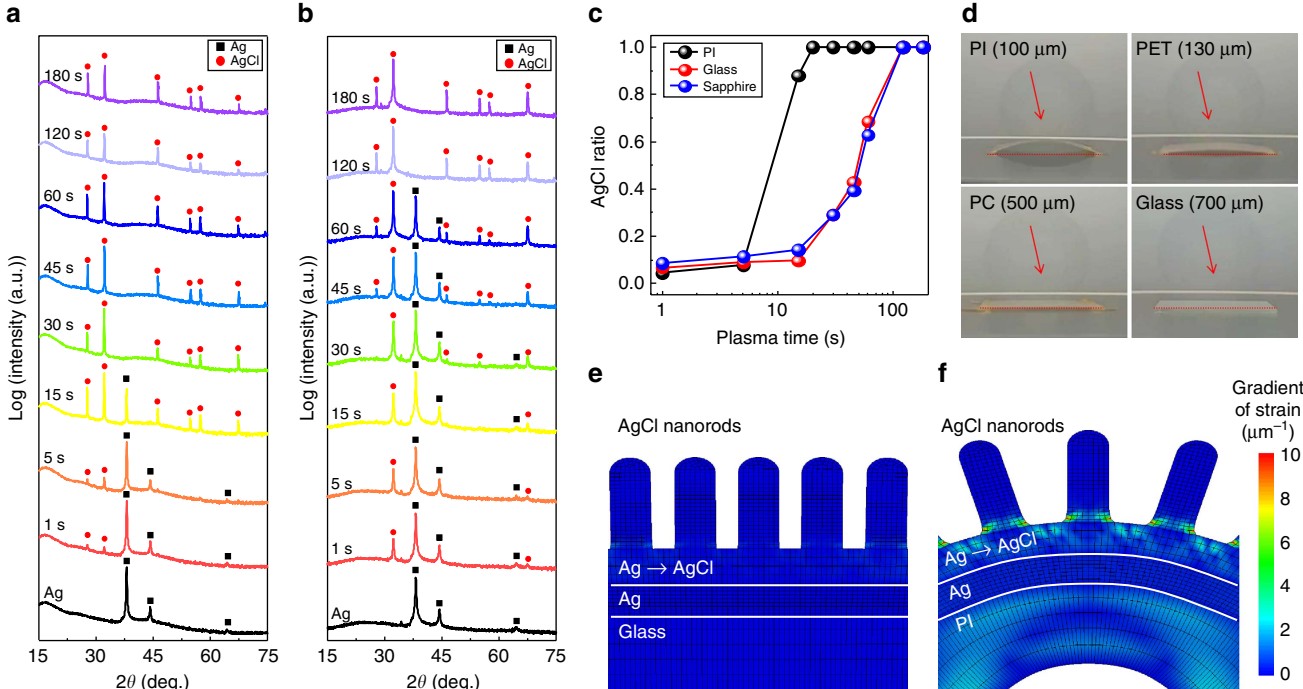

**Figure 7 | Effect of substrate flexibility on growth of AgCl nanorods.** X-ray diffraction (XRD) patterns of $Cl_2$-exposed Ag film as a function of exposure time on (**a**) polyimide and (**b**) soda-lime glass; Ag (black squares); AgCl (red circles). (**c**) Partial AgCl ratio calculated from XRD patterns. (**d**) Photography of Ag-coated 100-μm-thick polyimide (PI), 130-μm-thick polyethylene terephthalate (PET), 500-μm-thick polycarbonate (PC) films and 700-μm-thick soda lime glass during $Cl_2$ exposure. Calculated gradient of strain under volume expansion from Ag to AgCl on (**e**) rigid glass and (**f**) flexible PI film. Volume expansion was assumed to occur at interface of AgCl nanorods with Ag film. In flexible PI film, strain was localized at bottoms of AgCl nanorods.

**The growth mechanism of AgCl NRs on flexible film**. The AgCl NRs grew rapidly as the substrate became thin (Fig. 2). To investigate the dependence of growth characteristics on the kind of substrate, XRD was measured on Ag-deposited PI film (Fig. 7a), glass (Fig. 7b) and c-sapphire (Supplementary Fig. 6) as a function of plasma time. As-deposited Ag film showed only metallic Ag peaks. As the plasma exposure time increased, AgCl peaks began to appear and the Ag peaks weakened. The ratio of AgCl (Fig. 7c) (that is, [area of AgCl peak]/[total areas of AgCl and Ag peaks]) reached 1.0 within 20 s on the PI film but it took 120 s on both glass and on c-sapphire; that is, AgCl grew six times faster on PI film than on the glass and on the c-sapphire. This fast growth on PI is due to the spontaneous bending with convex shape during the $Cl_2$ plasma treatment (Fig. 7d). During NR growth, the flexible thin PI could bend to minimize the strain energy induced by lattice mismatch between Ag and AgCl, but the inflexible thick polycarbonate (PC) film and glass did not bend. This difference occurs because the PI film has the smallest bending radius due to its low thickness ($d$) and low Young's modulus ($E$) from the Stoney equation[50] (PI film: $d_{PI} = 100\,\mu m$, $E_{PI} = 2.5\,GPa$; glass: $d_{glass} = 700\,\mu m$, $E_{glass} = 74\,GPa$). The details are described in Supplementary Table 1.

To further investigate the effect of substrate bending on the growth of NRs, we conducted finite element method analysis on rigid glass (Fig. 7e) and flexible PI film (Fig. 7f). The reaction of Ag with Cl causes to expand the volume of Ag layer because $a_{AgCl} = 5.545\,Å$ is greater than $a_{Ag} = 4.079\,Å$. This causes convex bending of PI film, but not of the stiff glass. The bending relaxes the strain, so flexion of the PI film caused strain distribution to vary rapidly at the Ag/AgCl interface (Supplementary Fig. 7b) because the bending relaxes the strain. As a result, the strain gradient of $11.0\,\mu m^{-1}$ was localized near the edge of the NRs in the PI film (Fig. 7f). However, in the case of glass, there was no strain gradient (Supplementary Fig. 7a) because of the high stiffness of the glass. The strain gradient on the glass was calculated to be only $0.79\,\mu m^{-1}$ (Fig. 7e). The strain gradient was calculated on the various substrates such as PI, PET, PC film, glass and sapphire. The elastic properties of substrates are summarized in Supplementary Table 1. The bending radius of substrates were calculated using the Stoney equation[51] and normalized to that of PI (Supplementary Table 1). The strain gradient at the edge of NRs was plotted as a function of relative bending radius (Supplementary Fig. 8). As the degree of convex flexion increased, the strain gradient became increasingly localized at the edge of the NRs. The localization of strain gradient can increase the strain-induced diffusion of Ag atoms as[25,52]

$$J = -D\nabla C - \frac{C\Omega DE}{k_B T}\nabla\varepsilon \approx -\frac{C\Omega DE}{k_B T}\nabla\varepsilon,$$

where $J$ is the atomic flux, $C$ is the atomic concentration, $\Omega$ is the atomic volume, $D$ is the local diffusion coefficient, $E$ is Young's modulus, $k_B$ is Boltzmann's constant, $T$ is absolute temperature and $\varepsilon$ is the strain. In general, strain-induced diffusion is more dominant than that induced by the concentration gradient at low temperatures[53], so the concentration gradient was neglected (Supplementary Table 2). Because the strain gradient was localized at the edge of the NRs on the PI film, strain-induced diffusion drove facilitated diffusion of Ag atoms toward the bases of the NRs, so NRs growth was accelerated on the PI film.

To test this hypothesized mechanism of rapid growth on PI, we attached the Ag-coated PI film to a rigid substrate (that is, the sample chuck in the plasma chamber) using double-sided kapton tape, so that the substrate could not bend under lattice-mismatch strain. Then the sample was exposed to $Cl_2$ plasma for 45 s (Fig. 8). It was found that no NRs were grown on the fixed PI film, as in the case of the rigid glass substrate. On the basis of this experiment, we

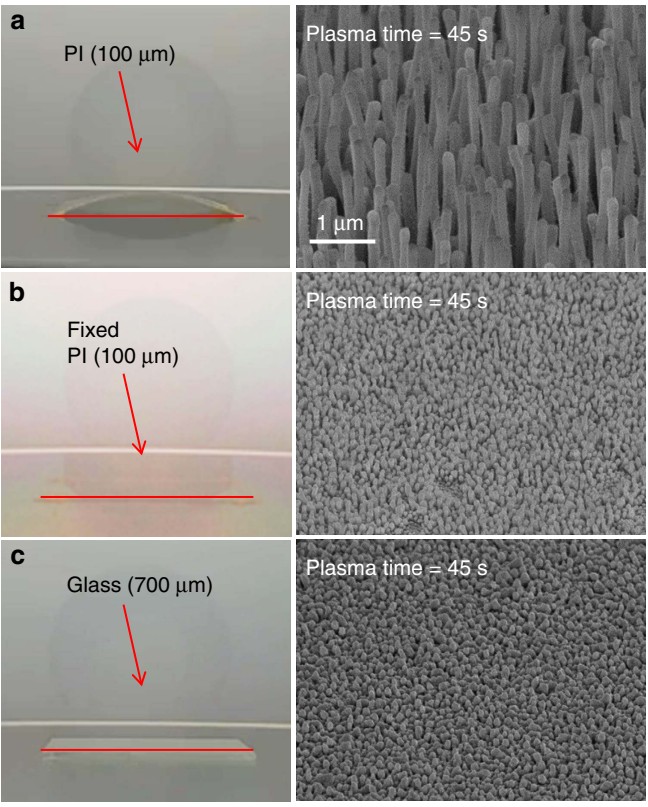

**Figure 8 | Effect of bending of substrate on growth of nanorods.** 300-nm-thick Ag coated on (**a**) polyimide (PI) film, (**b**) inflexible PI film fixed by tape and (**c**) glass were exposed $Cl_2$ plasma for 45 s. (Left) Photographs of the substrates during $Cl_2$ plasma exposure. (Right) Scanning electron microscopy images after plasma treatment.

conclude that the substrate bending critically contributes to the rapid growth of NRs, due to strain-induced diffusion. Experimental results provide evidence that Ag atoms diffuse to the bases of the NRs and react with Cl radicals until Ag is completely consumed, leading to the formation of single-crystal AgCl NRs.

**Application of the AgCl NRs in OLEDs**. These AgCl NRs could be applied to OLEDs. To investigate the effect of AgCl NRs on light extraction of the OLEDs, AgCl NRs on the backside of PI film were used as substrates. The device structure was $WO_3$ (36 nm)/Ag (12 nm)/$WO_3$ (3 nm)/$WO_3$:CBP (230 nm)/CBP (10 nm)/Ir(ppy)$_3$:CBP (30 nm)/TPBi (45 nm)/LiF (1 nm)/Al (100 nm). The reference device was fabricated on as-received PI film without AgCl nanostructure. The emission wavelength of Ir(ppy)$_3$ was 510 nm. To enable investigation of the effects of AgCl NRs on antireflection and light scattering, the device was designed to trap a large amount of light.

The current density–voltage ($J$–$V$) characteristics of OLEDs based on PI film with AgCl NRs are given in Fig. 9a. The measured operating voltages of a reference device and AgCl device (AgCl on the back of PI film) were ranged from 7.7 to 8.0 V at $J = 25$ mA cm$^{-2}$; the difference is negligible. The current efficiency vs. luminance of the OLEDs (Fig. 9b) increased as $t_{Ag}$ increased to 200 nm, but decreased at $t_{Ag} = 300$ nm. At 1,000 cd m$^{-2}$, the maximum current efficiency was 50.3 cd A$^{-1}$ in the reference device, 54.8 cd A$^{-1}$ at $t_{Ag} = 10$ nm, (8.9% increase vs. reference) and 66.9 cd A$^{-1}$ at $t_{Ag} = 200$ nm (33.0% increase vs. reference). The increase of current efficiency at $t_{Ag} = 10$ nm originated from increased $T_{total}$ due to the refractive index grading effect provided

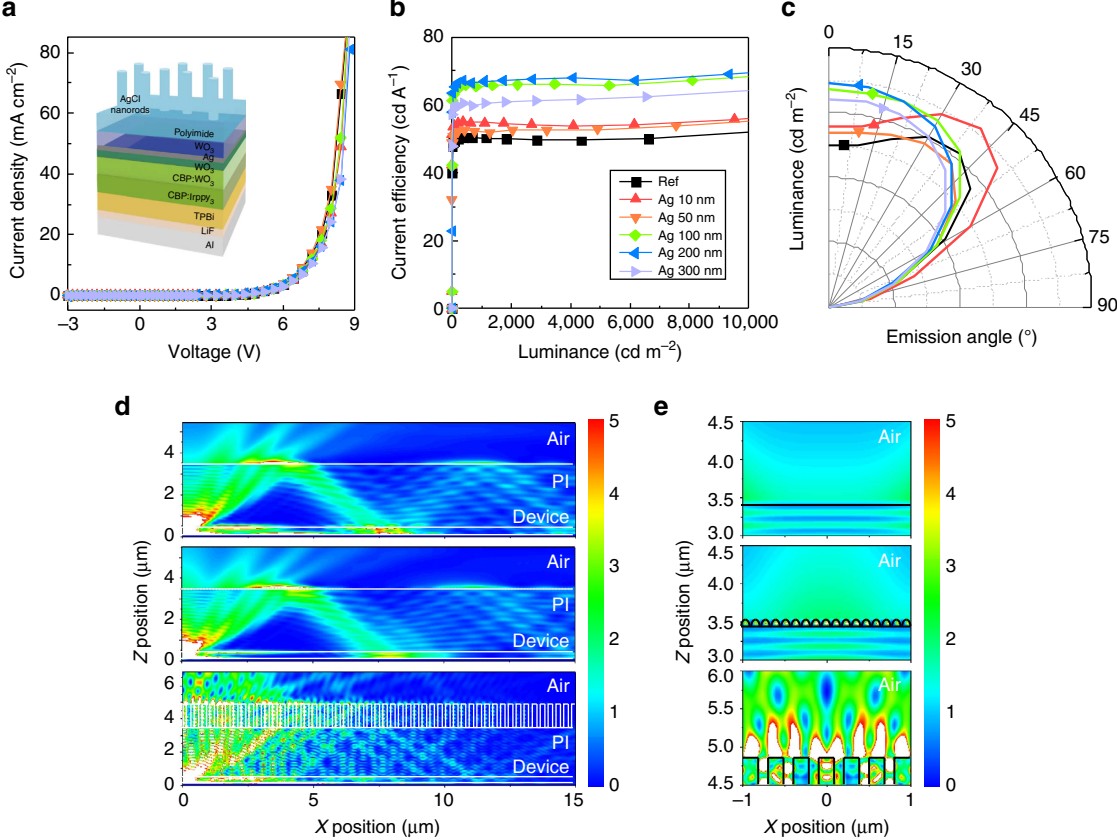

**Figure 9 | Characterization of the AgCl nanorods implemented organic light-emitting diodes (OLEDs).** (**a**) Current density–voltage (*J–V*) characteristics of OLEDs with various AgCl nanostructures. (**b**) Current efficiency of various AgCl nanostructures. (**c**) Angle-resolved luminance with various AgCl nanostructures at current density of 25 mA cm$^{-2}$. Calculated cross-sectional electric field distribution of (**d**) flat device (top), device with AgCl obtained using 10-nm Ag (centre) and 200-nm Ag (bottom). (**e**) Enlarged view of the electric field distribution near interface between substrate and air.

by the sub-wavelength structure. In contrast, the increase in current efficiency at $t_{Ag} = 200$ nm was a result of increased *H*. The AgCl NRs could extract confined waveguide modes in the substrate by light scattering effects.

The angular dependence of emission characteristics of the devices varied with $t_{Ag}$ (Fig. 9c). The device with $t_{Ag} = 10$ nm showed increased emission intensity over a whole range of viewing angles without modifying the emission shape compared to the reference device. The haze-free nanostructure with $t_{Ag} = 10$ nm did not induce any light scattering, which can affect angular dependency of emission, so the emission pattern of this device was the same as that of the flat device. However, as $t_{Ag}$ increased, the angular emission patterns of AgCl device approached a Lambertian intensity distribution; the change was mainly due to strong scattering of light by AgCl NRs. The emission of the scattered light in all azimuthal directions could contribute to improved angular dependence of colour stability of the OLEDs. The Commission Internationale de l'éclairage (CIE) 1931 colour coordinates of the OLED with $t_{Ag} = 200$ nm did not change as emission angle increased (Supplementary Fig. 9). These results indicate that the subwavelength-scale AgCl obtained using $t_{Ag} = 10$ nm is suitable for high-definition display windows by increasing luminance without sacrificing optical clarity, whereas wavelength-scale AgCl obtained using $t_{Ag} = 200$ nm is suitable for light illumination by increasing light extraction efficiency and improving colour uniformity over all emission angles.

To theoretically study the optical effects of the AgCl NRs on the light extraction efficiency for the OLEDs, two-dimensional finite-difference time-domain (2D FDTD) simulations were

performed. Except at the bottom layer, the perfect matched layer was used as the boundary condition to avoid occurrence of an unwanted reflected wave at the edge of the structure. At the bottom layer, the perfect electric conductor boundary condition was used to approximate the metallic mirror at the bottom of the OLEDs. The vertical and horizontal point dipole sources were excited with a wavelength of 510 nm in the active layer. The simulations were performed until steady-state was reached (that is, change of electric field is < 0.01% per FDTD time step). The cross-sectional electric field distributions were obtained at steady state by using a discrete Fourier transform (DFT) monitor. The vertical and horizontal dipoles were incoherently combined to get the electric field distribution under an un-polarized dipole source.

The electric field distributions (Fig. 9d) were obtained for the flat device, and for devices with $t_{Ag} = 10$ nm and with $t_{Ag} = 200$ nm. In the flat device, the electromagnetic wave was confined in the substrate due to total internal reflection at the interface between air and the PI film. In the device with $t_{Ag} = 10$ nm, the electric field near the centre (*X* position = 0) was enhanced compared to that of the flat device (Fig. 9e), because sub-wavelength scale AgCl provided gradually changed refractive index. Because haze was zero, the field distribution in the air was similar to that on the flat device. However, the device with wavelength-scale (on $t_{Ag} = 200$ nm) AgCl NRs showed dynamic electric field intensity outside the PI film because the NRs extract the confined wave from the PI substrate, so the field intensity inside the PI film was significantly decreased in the device that had the AgCl NRs. As a result, the calculated light extraction efficiency of the OLED was 46% greater in the device with NRs

than in the flat device. Thus, AgCl NRs effectively increase light extraction by inducing light scattering.

The AgCl NRs also showed excellent mechanical flexibility. To verify the mechanical stability of the AgCl NRs on PI film, we conducted bending test with various bending radius and bending cycles. The NRs were stable as the bending radius ($r$) reduced to 1.73 mm (Fig. 10a). No changes in the morphology of NRs and no cracks were found after up to 10,000 bending cycle at $r = 1.73$ mm (Fig. 10b). Furthermore, the average total transmittance and haze of PI film were sustained even after 10,000 bending cycles (Fig. 10c).

## Discussion

We have developed single-crystalline AgCl NRs on various kinds of substrates. By only exposing Ag to $Cl_2$ plasma, we achieve vertically well-aligned NRs within 1 min, which is the fastest growth ever reported for the growth of single-crystalline NRs. On the basis of experimental observations, we proposed the following growth mechanism of the AgCl NRs, Ag atoms consistently diffuse to the base of NRs and react with incorporated Cl radicals to form AgCl. During the reaction, AgCl is subjected to compressive strain due to volume expansion from Ag to AgCl. During the strain relaxation process, AgCl NRs grow along the [200]-preferred direction on the (111) Ag film to minimize lattice mismatch. XRD reveals that interface relationship between AgCl (200) and Ag (111) is cube-on-hexagon epitaxy. We discussed the effect of bendability of substrate on growth: flexion of the substrate facilitates diffusion of Ag atoms to the base of NRs, thereby promoting their growth. The size of the AgCl nanostructure can be controlled from sub-wavelength scale to wavelength scale simply by changing the thickness of the Ag layer. As a result, light scattering was successfully controlled,

which is a critical requirement for nanostructure-based devices. The AgCl NRs on PI film were used in OLEDs. The sub-wavelength AgCl increased OLED luminance by 8.9% compared to as-received PI film, and did not induce blurring or haziness. Due to light scattering, wavelength-scale NRs improved luminance efficiency by 33.0% and reduced the change of colour uniformity over emission angle.

There were a number of reports on the growth of NR using several kinds of growth techniques. Inorganic nanowires can be synthesized by various methods such as VLS, CVD, thermal oxidation and hydrothermal (Supplementary Table 3). However, VLS, CVD and the thermal oxidation methods required high temperature (greater than 400 °C), which would damage a polymer substrate. The hydrothermal method can grow NRs at low temperatures (about 100 °C), but the growth rate is too slow (less than 1 nm min$^{-1}$). The high process temperatures and slow rates of the previous methods are serious disadvantages for application to polymer substrates. In this work, we found a way to use a $Cl_2$ plasma source to grow the single-crystalline and vertically well-aligned AgCl NRs. The advantages of this technique are low temperature process (room temperature) and high-growth rate (about 2,000 nm min$^{-1}$). Such process condition allows the AgCl NRs to implement the R2R process shown in Fig. 1.

To confirm the applicability of the method to R2R process, we designed a virtual roll-to-roll (R2R) system (Supplementary Fig. 10). The plasma exposure time is a critical factor for application to R2R process. The plasma time of 45 s is sufficient to get the NRs (Fig. 2). Because the deposition rate of Ag layer did not have a significant effect on morphology or growth rate of AgCl NRs (Supplementary Fig. 11), the deposition rate was set to be 20 Å s$^{-1}$.

When the plasma chamber was 90 cm long, and the Ag-deposition chamber was 300 cm long, the feed rate of R2R

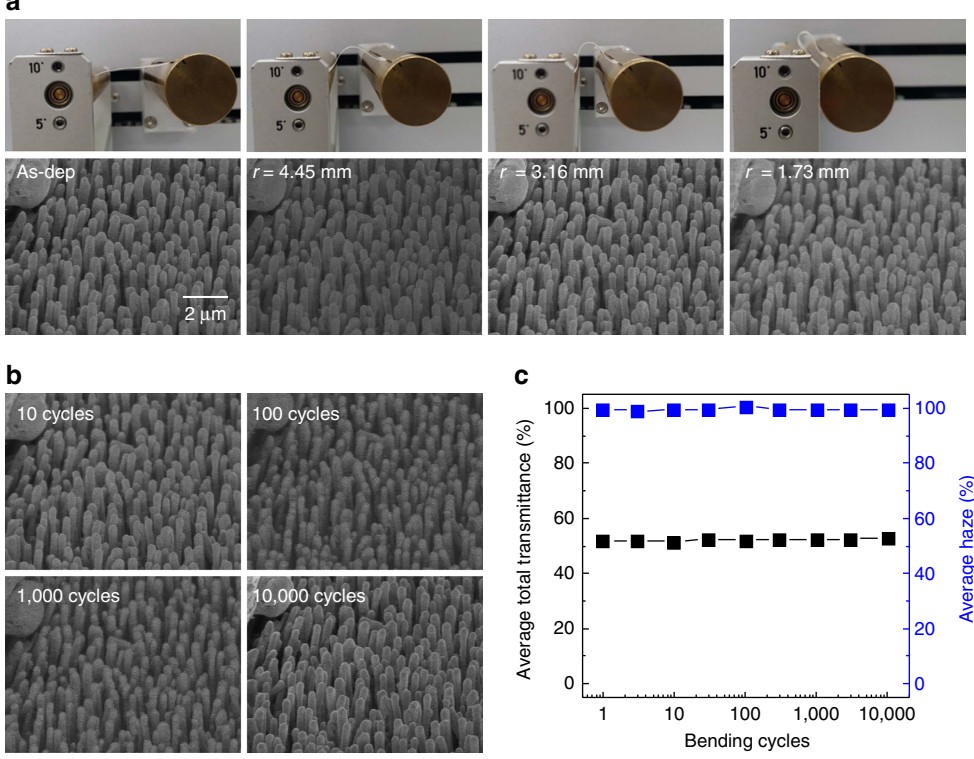

**Figure 10 | Mechanical stability of the AgCl nanorods. (a)** Photographs (top) and Scanning electron microscopy (SEM) images (bottom) of AgCl nanorods (NRs) on polyimide (PI) film with bending radius ($r$). (**b**) SEM images of AgCl NRs on PI film with bending cycle at $r = 1.73$ mm. Same position was observed using mark at upper left corner. (**c**) Average total transmittance and haze of AgCl NRs as a function of bending cycles at $r = 1.73$ mm.

process was $1.2 \, \text{m min}^{-1}$, which could be used in a commercial process. As a result, the plasma-induced AgCl NRs could be rapidly fabricated using an R2R process over a large area on several kinds of polymer film because the method does not require lithography, mask moulds or thermal processes. These size-tunable and plasma-induced AgCl NRs can promote the commercialization of highly efficient, inexpensive flexible optoelectronic devices.

## Methods

**Fabrication of AgCl NRs.** The colourless PI films were ultrasonically cleaned with isopropyl alchol for 5 min, then dried under blowing $N_2$ gas and baked at 110 °C for 10 min. The Ag layer was deposited at $1 \, \text{Å s}^{-1}$ on the PI film by using an e-beam evaporator under $1 \times 10^{-6}$ Torr. Then PI films with Ag layers were treated using $Cl_2$ plasma. The plasma was induced using 350-W RF power in $Cl_2$ ambient. The chamber pressure was maintained at 10 mTorr during treatment.

**Characterized instrument.** The total and diffused transmittance of the AgCl/PI sample were measured using a ultraviolet–visible–NIR spectrophotometer (Cary 4000, Agilent) with a diffuse reflectance accessory. The surface morphology of the AgCl was measured using a field-emission scanning electron microscope (XL30S FEG, PHILIPS) with an 5 kV acceleration voltage and 6 mm working distance, and an atomic force microscope operated in tapping mode on a Veeco nanoscope III. XPS was measured using a synchrotron radiation photoemission spectroscopy with the incident X-ray source at 650 eV and a base pressure of $5 \times 10^{-10}$ Torr at the 4D beam line at the Pohang Accelerator Laboratory (PAL). The chemical bonding states were separated using a combination of Gaussian and Lorentzian functions. The XRD patterns were measured using a house X-ray source (D2 phaser, Bruker) at 30 kV and 10 mA. Off-axis phi scans using synchrotron radiation were performed at the 3D beamline at PAL. The HRTEM (JEM 2200FS, JEOL) was conducted at 200 kV with a Cs-corrector. The AgCl NRs were sonicated in ethanol and put in a Cu grid. To prevent e-beam decomposition of the AgCl, 10 nm of carbon was deposited on both sides of the Cu grid.

**Fabrication and characterization of OLEDs.** To investigate how AgCl NRs affect OLED properties, conventional bottom-emission OLEDs were fabricated. The following layers were deposited using a thermal evaporator under a base pressure of 1 μTorr. A dielectric ($WO_3$ 36 nm)/metal (Ag 12 nm)/dielectric ($WO_3$ 3 nm) structure was used for anode electrodes. Then 4,4′-N,N′-dicarbazole-biphenyl (CBP) doped with 15% $WO_3$ as a 230-nm hole injection layer (HIL), intrinsic CBP as a 10-nm hole transport layer (HTL), Tris[2-phenylpyridinato-C2, N]iridium(III) Ir(ppy)$_3$ doped CBP as a 30-nm emissive layer (EML), 1,3,5-Tris(1-phenyl-1H-benzimidazol-2-yl)benzene (TPBi) as a 45-nm hole blocking layer (HBL), LiF as a 1-nm electron injection layer (EIL) and Al as a 100-nm reflective cathode were deposited in sequence.

The $J–V$ characteristics of the OLEDs were measured using a Kiethley 2400 source metre in $N_2$ ambient. Luminance characteristics of the OLEDs were measured using a calibrated colorimeter (Qbism HEXA50, McScience) in $N_2$ atmosphere. The EL spectra of OLEDs were measured using a spectrometer (SM240, Spectral Products) equipped with an optical fibre.

**Theoretical modelling and calculation.** Two-dimensional finite element method analysis were performed to investigate the effect of bending on growth of AgCl NRs. The simulation tool was commercial software ABAQUS v6.9. The simulation structure consisted of AgCl NRs ($E = 20 \, \text{GPa}$, $v = 0.4$), Ag ($E = 83 \, \text{GPa}$, $v = 0.37$) 150 nm, and PI ($E = 2.5 \, \text{GPa}$, $v = 0.34$) or glass ($E = 74 \, \text{GPa}$, $v = 0.3$) substrates, which represented the intermediate state of NR growth. The thickness of the substrate was reduced to 500 nm. On the glass, the bottom of the structure was pinned to represent the rigid glass. The AgCl NRs were set to have period = 300 nm, diameter = 100 nm and height = 600 nm. The reaction of Ag with Cl was assumed to cause volume expansion at the interface of NRs with Ag film. Thus, the top of Ag layer was expanded by 36% and the elastic property of the layer changed from Ag to AgCl. The mesh size of the substrates was 50 nm and the sizes of Ag and AgCl were 25 nm.

Two-dimensional FDTD method with a perfect matched layer was used for numerical analysis of light extraction. The simulation tool was electromagnetic caclulation module (R Soft Fullwave) from commercial software R soft 2014. 09 (Synopsys and R soft Design Group, Inc.) The simulation structure consists of an Aluminium cathode (100 nm), LiF (refractive index $n = 1.39$) 1 nm, TPBI ($n = 1.75$) 45 nm, CBP:Ir(ppy)$_3$ ($n = 1.75$) 30 nm, CBP ($n = 1.75$) 10 nm, CBP:$WO_3$ ($n = 1.77$) 230 nm, $WO_3$ ($n = 1.92$) 3 nm, Ag 12 nm, $WO_3$ ($n = 1.92$) 36 nm and PI substrate ($n = 1.6$) 3 μm. The thickness of the PI substrate was reduced to 3 μm larger than five times the optical wavelength, because the real thickness of PI substrate (100 μm) is too large and consumed too much time in the FDTD simulation. The total width of the structure is 30 μm, which is enough to get saturated extraction efficiency as a function of width. On the basis of analysis of SEM and AFM images, the sub-wavelength AgCl nanostructure was set to have period = 60 nm, diameter = 50 nm and height = 70 nm; and the wavelength-scale

AgCl NR was set to have period = 300 nm, diameter = 200 nm and height = 1.5 μm The grid size of the simulation was 10 nm, which is small enough to resolve AgCl nanostructures. Due to this large calculation domain and small grid size, 2D simulation was used to reduce computation time. Horizontal and vertical dipole sources were used. The DFT monitor was used to get the cross-sectional spatial electric field distribution during steady state.

**Data availability.** The data that support the findings of this study are available from the corresponding author upon request.

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

## Acknowledgements

This research was financially supported in part by the National Research Foundation (NRF) Grant funded by the Korean Government (NRF-2014H1A2A1021655-Global PhD. Fellowship Program).

## Author contributions

J.-L.L. supervised the project, analysed all the data and prepared the manuscript. J.Y.P. performed most of experiments, numerical simulation and manuscript preparation. I.L. conceived the idea and design the project. J.H. contributed to interpretation of experimental data and organization of the paper. S.G. contributed to the fabrication of OLED devices with AgCl nanorods. All authors discussed the results and commented on the manuscript.

## Additional information

**Competing interests:** The authors declare no competing financial interests.

