## [Peer Review File · Nature Communications]

Reviewers' comments:

Reviewer #1 (Remarks to the Author):

The main concept of the paper is to grow AgCl nanorod arrays on flexible substrates for potential application in future devices. The AgCl nanorods are found to grow epitaxially on the silver film which was first deposited on the substrate. The novelty of the Cl plasma assisted process is one of low temperature which is compatible with plastic substrates.

A model is developed to explain the observed growth process—strained induced diffusion of Ag atoms as the driving force of the nanorod growth. This takes place at the interface between the Ag film and the initial AgCl nano-particles. It is difficult to compare the measured rate of nano-rod growth with the model. However, such a model sounds reasonable.

Using the nanorods to improve the coupling of light into and out of a device such as LED is not new. In fact, one can optimize the array geometry to optimize the range of light wavelengths for the desired coupling.

The publication of this paper should be considered. The authors should include following:

1. Can one flex the nanorod array on the plastic substrate once it is fabricated?
2. Why is the choice of AgCl nanorods unique? For example, what about ZnO nanorods?
3. How can one verify experimentally the strain-relaxation driven growth mechanism as proposed in Figure 7?

Reviewer #2 (Remarks to the Author):

The authors of this manuscript described the preparation of single-crystalline AgCl nanorods by using a Cl₂ plasma source. A growth mechanism based on a strain-relaxation process was proposed.

The manuscript is well written and the topic is of interest. However, I would recommend submission to a lower impact journal as the novelty of the work does not justify publication in Nature Communications –impact factor 11.329.

Reviewer #3 (Remarks to the Author):

park et al report a rigorous methodology for the controlled growth of AgCl nanorods on a range of substrate materials using a Cl₂ plasma source. The authors describe the influence on the nano rods of various growth parameters and provide details (and accurate) statistical information probing the nucleation, evolution and growth of the nano rods.

The data is generally clearly presented and easy to interpret however some of the claims are extraordinarily inaccurate - it is unclear whether this is a fault of the authors or an issue with the use of English. For example on more than one occasion the authors claim deposition was achieved on 'any kind of substrate' (P3L12) and again "were grown on any substrate" (P3L22) - clearly only a few transparent (excluding sapphire for growth moddelin) substates were used so i suggest this is canned to something like " a range transparent substrates were used, including rigid (glass) and flexible (Pet...)".

The poor use of English detracts from the communication in the manuscript e.g. "Cl radicals react

consistently"(abstract), nanostructure vs nanostructure (entire manuscript), "flexible plastic substrates are flexible..."(P2L1), "higher power efficiency than did a device" (P2last line)...however quality improves as the documents extends. One recurring issue is the use of "chlorized", which i assume is from silanized? this is improper and should be replaced with "completely reacted" or similar.

Finally the applicability of the processing on flexible/r-2-r substrates is interesting but this is e.g. Fig 1 somewhat ambitious given the nature of the dew;;/processing time and the feed rate typical with such systems. The reader should be able to infer this application without it being specified.

Specific comments

Figures all - please remove the system or secondary scale bars on images, this is distracting and messy

Fig 6 and S6 - comment required in text about the emergence of the (111)(311)(222) peaks and how this fits the growth model proposed is required in the text.

Fig S4 - overall quality of image a is very poor

Fig 1 - see proviso comments, I'm not sure a r-2-r system is good use of a figure and have to question the immediate suitability of the growth method and this technique.

Fig S1 shows an XRD pattern, not patterns. The line width of the plot hides key features of the diffraction peaks and should be reduced.

Fig 3a - should be annotated to include the Ag thickness of the substrate.

I believe that if these comments were addressed that the manuscript could be considered for publication.

Replies on reviewer's comments

We appreciate very much for your kind comments on our paper. We have responded point by point to the comments of the reviewers and revised the manuscript in accordance with the reviewers' comments. English was improved through a professional editorial service for grammatical correction. Corrected words and phrases are highlighted with red color through the manuscript to be recognized easily.

Replies on reviewer's comments are given below.

<For Reviewer #1>

The main concept of the paper is to grow AgCl nanorod arrays on flexible substrates for potential application in future devices. The AgCl nanorods are found to grow epitaxially on the silver film which was first deposited on the substrate. The novelty of the Cl plasma assisted process is one of low temperature which is compatible with plastic substrates.

A model is developed to explain the observed growth process—strained induced diffusion of Ag atoms as the driving force of the nanorod growth. This takes place at the interface between the Ag film and the initial AgCl nano-particles. It is difficult to compare the measured rate of nanorod growth with the model. However, such a model sounds reasonable.

Using the nanorods to improve the coupling of light into and out of a device such as LED is not new. In fact, one can optimize the array geometry to optimize the range of light wavelengths for the desired coupling.

The publication of this paper should be considered. The authors should include following:

1. Can one flex the nanorod array on the plastic substrate once it is fabricated?

(A) To verify the mechanical stability of the AgCl NRs on PI film, we conducted bending test with various bending radius and bending cycles. The NRs were stable as the bending radius (r) reduced to 1.73 mm (Fig. A1a). No changes in the morphology of NRs and no cracks were found after up to 10,000 bending cycle at $r = 1.73$ mm (Fig. A1b). Furthermore, the average total transmittance and haze of PI film sustained even after 10,000 bending cycles (Fig. A1c).

Figure A1. (a) Photographs (top) and SEM images (bottom) of AgCl NRs on PI film with bending radius (r). (b) SEM images of AgCl NRs on PI film with bending cycle at $r = 1.73$ mm. Same position was observed using mark at left upper corner. (c) Average total transmittance and haze of AgCl NRs as a function of bending cycles at $r = 1.73$ mm.

We added a paragraph and figures (Fig. 10) in the revised manuscript (Page 13, line 5)

→ The AgCl NRs also showed excellent mechanical flexibility. To verify the mechanical stability of the AgCl NRs on PI film, we conducted bending test with various bending radius and bending cycles. The NRs were stable as the bending radius (r) reduced to 1.73 mm (Fig. 10a). No changes in the morphology of NRs and no cracks were found after up to 10,000 bending cycle at $r = 1.73$ mm (Fig. 10b). Furthermore, the average total transmittance and haze of PI film sustained even after 10,000 bending cycles (Fig. 10c).

Figure 10. Mechanical stability of the AgCl nanorods. (a) Photographs (top) and SEM images (bottom) of AgCl NRs on PI film with bending radius (r). (b) SEM images of AgCl NRs on PI film with bending cycle at $r = 1.73$ mm. Same position was observed using mark at left upper corner. (c) Average total transmittance and haze of AgCl NRs as a function of bending cycles at $r = 1.73$ mm.

2. Why is the choice of AgCl nanorods unique? For example, what about ZnO nanorods?

(A) There were a number of reports on the growth of nanorod using several kinds of growth techniques. Inorganic nanowires could be synthesized by various methods such as vapor-liquid-solid (VLS), chemical vapor deposition (CVD), thermal oxidation and hydrothermal (Table. A1). However, the VLS, the CVD and the thermal oxidation methods required high temperature process (>400 °C), which damage a polymer substrate. The hydrothermal method can grow NRs at low temperatures (~ 100 °C), but the growth rate is too slow (< 1 nm/min). In short, the previously proposed growth methods have disadvantages for application to polymer substrates due to high temperature process (>150 °C), slow growth rate (\sim few nm/min) and a limited size of substrate. In this work, we found a way to use a Cl_2 plasma source to grow the single-crystalline and vertically well-aligned AgCl nanorods (NRs). The advantages of this technique are low temperature process (room temperature) and high growth rate ($\sim 2,000$ nm/min). As a result, the AgCl nanorods could be applicable to flexible polymer substrates and we confirmed the size-tunable AgCl nanorods from sub-wavelength scale (<400 nm) to wavelength scale (>400 nm) by adjusting the thickness of Ag. Thus, optical behavior could managed from near-zero haze (0.23%) to full-scattering (100 %). The organic light emitting diodes with the wavelength scale AgCl nanorods showed a remarkable enhancement in luminance efficiency (66.9 cd/A at 1,000 cd/m^2) up to 33%, compared to that of reference (50.3 cd/A at 1,000 cd/m^2). The wavelength-scale nanorods were very effective in extracting the confined wave-guided electromagnetic wave in the substrate.

We extend the plasma process to the fabrication of ZnO nanorods. Zn metal was deposited at 1 Å/s on the PI film by using thermal evaporator under 1×10^{-5} Torr. Then, the Zn-coated PI film was treated for 45 s using O_2 plasma. The plasma was induced using 350-W RF power in O_2 ambient. The chamber pressure was maintained at 10 mTorr during treatment. No nanorods were produced (Fig. A2). And we found that the Zn film is forming a platy structure independent of O_2 plasma. However we believe that there is a room to produce the nanorods by optimizing the process condition.

Table A1. Growth temperature and rate of various inorganic nanorods

Growth Method	Materials	Growth Temperature [°C]	Growth Rate [nm/min]	Ref.
Hydrothermal	ZnO	90	0.6	1
	ZnO	60	1	2
	ZnO	95	6.25	3
	ZnO	70	2.9	4
	ZnO	79	11.4	5

	ZnO	90	5	6
	ZnO	90	6	7
	ZnO	100	2.8	8
	TiO ₂	140	3.7	9
	TiO ₂	150	1.2	10
	TiO ₂	90	17.1	11
	TiO ₂	150	5.6	12
	TiO ₂	170	0.23	13
	TiO ₂	190	23	14
	TiO ₂	210	10	15
	TiO ₂	200	2.08	16
	Ta ₂ O ₅	240	0.41	17
	WO ₃	170	7.5	18
	WO ₃	180	6.25	19
	Nb ₂ O ₅	150	4	20
	Fe ₂ O ₃	60	1.3	21
	CoO	120	17	22
Vapor-liquid-solid	ITO	300	30	23
	ITO	800	16.3	24
	ZnO	990	100	25
	ZnO	820	230	26
	ZnO	750	15	27
	NiO	550	30	28
	MgO	925	1.7	29
	SnO ₂	750	5	27
	In ₂ O ₃	750	15	27
	In ₂ O ₃	750	-	30
MgO	750	17	27	
Chemical Vapor Deposition	ZnO	650	90	31
	TiO ₂	550	-	32
	WO ₃	500	61	33
	WO ₃	500	108	34
	CuO	600	900	35
	SnO ₂	760	55.5	36
	Ga ₂ O ₃	1000	120	37

Thermal oxidation	ZnO	500	33.3	38
	CuO	400	13.8	39
	CuO	400	5.4	40
	CuO	400	14	41
	CuO	600	9.7	42
	CuO	400	11.1	43
	CuO	400	20	44
	W ₁₈ O ₄₉	700	12	45
AgCl	This work	~ RT	2000	

1. Cheng J. J., Nicaise S. M., Berggren K. K., Gradecak S. Dimensional tailoring of hydrothermally grown zinc oxide nanowire arrays. *Nano Lett.* **16**, 753-759 (2015).
2. Joo J., Chow B. Y., Prakash M., Boyden E. S., Jacobson J. M. Face-selective electrostatic control of hydrothermal zinc oxide nanowire synthesis. *Nat. Mater.* **10**, 596-601 (2011).
3. Park G. C., *et al.* Hydrothermally grown In-doped ZnO nanorods on p-GaN films for color-tunable heterojunction light-emitting-diodes. *Sci. Rep.* **5**, 10410 (2015).
4. Lee J. M., No Y.-S., Kim S., Park H.-G., Park W. I. Strong interactive growth behaviours in solution-phase synthesis of three-dimensional metal oxide nanostructures. *Nat. Commun.* **6**, 6325 (2015).
5. Watanabe K., *et al.* Arbitrary cross-section SEM-cathodoluminescence imaging of growth sectors and local carrier concentrations within micro-sampled semiconductor nanorods. *Nat. Commun.* **7**, 10609 (2016).
6. Kim B. H., Kwon J. W. Metal catalyst for low-temperature growth of controlled zinc oxide nanowires on arbitrary substrates. *Sci. Rep.* **4**, 4379 (2014).
7. Consonni V., *et al.* Selective area growth of well-ordered ZnO nanowire arrays with controllable polarity. *ACS nano* **8**, 4761-4770 (2014).
8. Yue H. Y., *et al.* ZnO nanowire arrays on 3D hierarchical graphene foam: biomarker detection of Parkinson's disease. *ACS nano* **8**, 1639-1646 (2014).
9. Ye M., Liu H. Y., Lin C., Lin Z. Hierarchical Rutile TiO₂ Flower Cluster-Based High Efficiency Dye-Sensitized Solar Cells via Direct Hydrothermal Growth on Conducting Substrates. *Small* **9**, 312-321 (2013).
10. Huang H., *et al.* Hydrothermal Growth of TiO₂ Nanorod Arrays and In Situ Conversion to Nanotube Arrays for Highly Efficient Quantum Dot-Sensitized Solar Cells. *Small* **9**, 3153-3160 (2013).
11. Yang T., *et al.* Position-controlled hydrothermal growth of periodic individual ZnO nanorod arrays on indium tin oxide substrate. *J. Phys. Chem. C* **118**, 20613-20619 (2014).
12. Berhe S. A., Nag S., Molinets Z., Youngblood W. J. Influence of Seeding and Bath Conditions in

- Hydrothermal Growth of Very Thin (~ 20 nm) Single-Crystalline Rutile TiO₂ Nanorod Films. *ACS Appl. Mater. Interfaces* **5**, 1181-1185 (2013).
13. Chen J., Yang H. B., Miao J., Wang H.-Y., Liu B. Thermodynamically driven one-dimensional evolution of anatase TiO₂ nanorods: one-step hydrothermal synthesis for emerging intrinsic superiority of dimensionality. *J. Am. Chem. Soc.* **136**, 15310-15318 (2014).
 14. Wu W. Q., Huang F., Chen D., Cheng Y. B., Caruso R. A. Thin Films of Dendritic Anatase Titania Nanowires Enable Effective Hole-Blocking and Efficient Light-Harvesting for High-Performance Mesoscopic Perovskite Solar Cells. *Adv. Funct. Mater.* **25**, 3264-3272 (2015).
 15. Resasco J., Dasgupta N. P., Rosell J. R., Guo J., Yang P. Uniform doping of metal oxide nanowires using solid state diffusion. *J. Am. Chem. Soc.* **136**, 10521-10526 (2014).
 16. Wang C.-C., Hsueh Y.-C., Su C.-Y., Kei C.-C., Perng T.-P. Deposition of uniform Pt nanoparticles with controllable size on TiO₂-based nanowires by atomic layer deposition and their photocatalytic properties. *Nanotechnology* **26**, 254002 (2015).
 17. Su Z., Wang L., Grigorescu S., Lee K., Schmuki P. Hydrothermal growth of highly oriented single crystalline Ta₂O₅ nanorod arrays and their conversion to Ta₃N₅ for efficient solar driven water splitting. *Chem. Commun.* **50**, 15561-15564 (2014).
 18. Zheng F., Lu H., Guo M., Zhang M., Zhen Q. Hydrothermal preparation of WO₃ nanorod array and ZnO nanosheet array composite structures on FTO substrates with enhanced photocatalytic properties. *Journal of Materials Chemistry C* **3**, 7612-7620 (2015).
 19. Zheng F., *et al.* Hydrothermal preparation, growth mechanism and supercapacitive properties of WO₃ nanorod arrays grown directly on a Cu substrate. *CrystEngComm* **18**, 3891-3904 (2016).
 20. He J., *et al.* Hydrothermal growth and optical properties of Nb₂O₅ nanorod arrays. *Journal of Materials Chemistry C* **2**, 8185-8190 (2014).
 21. Kong D., *et al.* Seed-assisted growth of α -Fe₂O₃ nanorod arrays on reduced graphene oxide: a superior anode for high-performance Li-ion and Na-ion batteries. *J. Mater. Chem. A* **4**, 11800-11811 (2016).
 22. Cao L., *et al.* Vertically aligned cobalt oxide nanowires on graphene networks for high-performance lithium storage. *Nanotechnology* **25**, 445704 (2014).
 23. Yu H. K., Lee J.-L. Growth mechanism of metal-oxide nanowires synthesized by electron beam evaporation: A self-catalytic vapor-liquid-solid process. *Sci. Rep.* **4**, 6589 (2014).
 24. Shen Y., *et al.* Epitaxy-Enabled Vapor-Liquid-Solid Growth of Tin-Doped Indium Oxide Nanowires with Controlled Orientations. *Nano Lett.* **14**, 4342-4351 (2014).
 25. Sallet V., Sartel C., Vilar C., Lusson A., Galtier P. Opposite crystal polarities observed in spontaneous and vapour-liquid-solid grown ZnO nanowires. *Appl. Phys. Lett.* **102**, 182103 (2013).
 26. Cheng G., *et al.* Large anelasticity and associated energy dissipation in single-crystalline nanowires.

- Nat. Nanotech.* **10**, 687-691 (2015).
27. Klamchuen A., *et al.* Rational Concept for Designing Vapor–Liquid–Solid Growth of Single Crystalline Metal Oxide Nanowires. *Nano Lett.* **15**, 6406-6412 (2015).
 28. Nagashima K., *et al.* Tailoring Nucleation at Two Interfaces Enables Single Crystalline NiO Nanowires via Vapor–Liquid–Solid Route. *ACS Appl. Mater. Interfaces* **8**, 27892-27899 (2016).
 29. Li L., Zhang X., Li L., Zhai X., Zeng C. Magnetoresistance of single-crystalline La_{0.67}Sr_{0.33}MnO₃/MgO nanorod arrays. *Solid State Commun.* **171**, 46-49 (2013).
 30. Domènech-Gil G., *et al.* Gas sensors based on individual indium oxide nanowire. *Sensors and Actuators B: Chemical* **238**, 447-454 (2017).
 31. Xu L., *et al.* Catalyst-free, selective growth of ZnO nanowires on SiO₂ by chemical vapor deposition for transfer-free fabrication of UV photodetectors. *ACS Appl. Mater. Interfaces* **7**, 20264-20271 (2015).
 32. Chen C., *et al.* Growth and characterization of well-aligned densely-packed rutile TiO₂ nanocrystals on sapphire substrates via metal–organic chemical vapor deposition. *Nanotechnology* **19**, 075611 (2008).
 33. Annanouch F. E., *et al.* Aerosol-assisted CVD-grown WO₃ nanoneedles decorated with copper oxide nanoparticles for the selective and humidity-resilient detection of H₂S. *ACS Appl. Mater. Interfaces* **7**, 6842-6851 (2015).
 34. Annanouch F. E., *et al.* Aerosol-assisted CVD-grown PdO nanoparticle-decorated tungsten oxide nanoneedles extremely sensitive and selective to hydrogen. *ACS Appl. Mater. Interfaces* **8**, 10413-10421 (2016).
 35. Lugo-Ruelas M., *et al.* Synthesis, microstructural characterization and optical properties of CuO nanorods and nanowires obtained by aerosol assisted CVD. *Journal of Alloys and Compounds* **643**, S46-S50 (2015).
 36. Deng K., Lu H., Shi Z., Liu Q., Li L. Flexible Three-Dimensional SnO₂ Nanowire Arrays: Atomic Layer Deposition-Assisted Synthesis, Excellent Photodetectors, and Field Emitters. *ACS Appl. Mater. Interfaces* **5**, 7845-7851 (2013).
 37. Hosein I. D., Hegde M., Jones P. D., Chirmanov V., Radovanovic P. V. Evolution of the faceting, morphology and aspect ratio of gallium oxide nanowires grown by vapor–solid deposition. *J. Cryst. Growth* **396**, 24-32 (2014).
 38. Zhao C., *et al.* Large-scale synthesis of bicrystalline ZnO nanowire arrays by thermal oxidation of zinc film: growth mechanism and high-performance field emission. *Cryst. Growth Des.* **13**, 2897-2905 (2013).
 39. Zhang Q., *et al.* Facile large-scale synthesis of vertically aligned CuO nanowires on nickel foam: Growth mechanism and remarkable electrochemical performance. *J. Mater. Chem. A* **2**, 3865-3874 (2014).

40. Rackauskas S., *et al.* In situ study of noncatalytic metal oxide nanowire growth. *Nano Lett.* **14**, 5810-5813 (2014).
41. Wang J., *et al.* Three-dimensional hierarchical Co₃O₄/CuO nanowire heterostructure arrays on nickel foam for high-performance lithium ion batteries. *Nano Energy* **6**, 19-26 (2014).
42. Li A., Song H., Zhou J., Chen X., Liu S. CuO nanowire growth on Cu₂O by in situ thermal oxidation in air. *CrystEngComm* **15**, 8559-8564 (2013).
43. Tang C., *et al.* Enhanced adhesion and field emission of CuO nanowires synthesized by simply modified thermal oxidation technique. *Nanotechnology* **27**, 395605 (2016).
44. Kargar A., *et al.* ZnO/CuO heterojunction branched nanowires for photoelectrochemical hydrogen generation. *ACS nano* **7**, 11112-11120 (2013).
45. Zhang Z., *et al.* Atomic-Scale Observation of Vapor–Solid Nanowire Growth via Oscillatory Mass Transport. *ACS nano* **10**, 763-769 (2015).

Figure A2. SEM images of (a) as-deposited Zn film on PI film and (b) O₂ plasma treated Zn film.

In order to emphasize the AgCl nanorods unique, we revised the manuscript as in below:

We added a sentence in the revised manuscript (page 3, line 9)

→ A polymer substrate flexes convexly during the growth, so migration of Ag atoms is accelerated by strain-induced diffusion, enabling the rapid growth of NRs on polymer substrate

We revised a sentence in the revised manuscript (page 4, line 15)

At a plasma time of 30 s, the nanodots began to grow as a shape of NRs, and their length gradually increased until 60 s.

→ At a plasma treatment time of 30 s, the nanodots began to grow to form a shape of NRs with length of 1 μm. Then, their length gradually increased until 60 s.

We added a paragraph in “Discussion” to emphasize the AgCl nanorods unique.

→ There were a number of reports on the growth of NR using several kinds of growth techniques. Inorganic nanowires can be synthesized by various methods such as vapor-liquid-solid (VLS), chemical vapor deposition (CVD), thermal oxidation and hydrothermal (Table. A1). However, the VLS, the CVD and the thermal oxidation methods required high temperature process (>400 °C), which would damage a polymer substrate. The hydrothermal method can grow NRs at low temperatures (~ 100 °C), but the growth rate is too slow (< 1 nm/min). In short, the previously proposed growth methods have disadvantages for application to polymer substrates due to high temperature process (>150 °C), slow growth rate (~few nm/min) and a limited size of substrate. In this work, we found a way to use a Cl₂ plasma source to grow the single-crystalline and vertically well-aligned AgCl. The advantages of this technique are low temperature process (room temperature) and high growth rate (~2,000 nm/min). Such process condition allows the AgCl NRs to implement the R2R process shown in Fig. 1.

We added Table S3 in the supplementary information of revised manuscript.

→ Supplementary Table S3. Growth temperature and rate of various inorganic nanorods

Growth Method	Materials	Growth Temperature [°C]	Growth Rate [nm/min]	Ref.
Hydrothermal	ZnO	90	0.6	1
	ZnO	60	1	2
	ZnO	95	6.25	3
	ZnO	70	2.9	4
	ZnO	79	11.4	5
	ZnO	90	5	6
	ZnO	90	6	7
	ZnO	100	2.8	8
	TiO ₂	140	3.7	9
	TiO ₂	150	1.2	10

	TiO ₂	90	17.1	11
	TiO ₂	150	5.6	12
	TiO ₂	170	0.23	13
	TiO ₂	190	23	14
	TiO ₂	210	10	15
	TiO ₂	200	2.08	16
	Ta ₂ O ₅	240	0.41	17
	WO ₃	170	7.5	18
	WO ₃	180	6.25	19
	Nb ₂ O ₅	150	4	20
	Fe ₂ O ₃	60	1.3	21
	CoO	120	17	22
Vapor-liquid-solid	ITO	300	30	23
	ITO	800	16.3	24
	ZnO	990	100	25
	ZnO	820	230	26
	ZnO	750	15	27
	NiO	550	30	28
	MgO	925	1.7	29
	SnO ₂	750	5	27
	In ₂ O ₃	750	15	27
	In ₂ O ₃	750	-	30
	MgO	750	17	27
Chemical Vapor Deposition	ZnO	650	90	31
	TiO ₂	550	-	32
	WO ₃	500	61	33
	WO ₃	500	108	34
	CuO	600	900	35
	SnO ₂	760	55.5	36
	Ga ₂ O ₃	1000	120	37
Thermal oxidation	ZnO	500	33.3	38
	CuO	400	13.8	39
	CuO	400	5.4	40
	CuO	400	14	41
	CuO	600	9.7	42

	CuO	400	11.1	43
	CuO	400	20	44
	W ₁₈ O ₄₉	700	12	45
AgCl	This work	~ RT	2000	

1. Cheng J. J., Nicaise S. M., Berggren K. K., Gradec̃ak S. Dimensional tailoring of hydrothermally grown zinc oxide nanowire arrays. *Nano Lett.* **16**, 753-759 (2015).
2. Joo J., Chow B. Y., Prakash M., Boyden E. S., Jacobson J. M. Face-selective electrostatic control of hydrothermal zinc oxide nanowire synthesis. *Nat. Mater.* **10**, 596-601 (2011).
3. Park G. C., *et al.* Hydrothermally grown In-doped ZnO nanorods on p-GaN films for color-tunable heterojunction light-emitting-diodes. *Sci. Rep.* **5**, 10410 (2015).
4. Lee J. M., No Y.-S., Kim S., Park H.-G., Park W. I. Strong interactive growth behaviours in solution-phase synthesis of three-dimensional metal oxide nanostructures. *Nat. commun.* **6**, 6325 (2015).
5. Watanabe K., *et al.* Arbitrary cross-section SEM-cathodoluminescence imaging of growth sectors and local carrier concentrations within micro-sampled semiconductor nanorods. *Nat. commun.* **7**, 10609 (2016).
6. Kim B. H., Kwon J. W. Metal catalyst for low-temperature growth of controlled zinc oxide nanowires on arbitrary substrates. *Sci. Rep.* **4**, 4379 (2014).
7. Consonni V., *et al.* Selective area growth of well-ordered ZnO nanowire arrays with controllable polarity. *ACS nano* **8**, 4761-4770 (2014).
8. Yue H. Y., *et al.* ZnO nanowire arrays on 3D hierarchical graphene foam: biomarker detection of Parkinson's disease. *ACS nano* **8**, 1639-1646 (2014).
9. Ye M., Liu H. Y., Lin C., Lin Z. Hierarchical Rutile TiO₂ Flower Cluster-Based High Efficiency Dye-Sensitized Solar Cells via Direct Hydrothermal Growth on Conducting Substrates. *Small* **9**, 312-321 (2013).
10. Huang H., *et al.* Hydrothermal Growth of TiO₂ Nanorod Arrays and In Situ Conversion to Nanotube Arrays for Highly Efficient Quantum Dot-Sensitized Solar Cells. *Small* **9**, 3153-3160 (2013).
11. Yang T., *et al.* Position-controlled hydrothermal growth of periodic individual ZnO nanorod arrays on indium tin oxide substrate. *J. Phys. Chem. C* **118**, 20613-20619 (2014).
12. Berhe S. A., Nag S., Molinets Z., Youngblood W. J. Influence of Seeding and Bath Conditions in Hydrothermal Growth of Very Thin (~ 20 nm) Single-Crystalline Rutile TiO₂ Nanorod Films. *ACS Appl. Mater. Interfaces* **5**, 1181-1185 (2013).
13. Chen J., Yang H. B., Miao J., Wang H.-Y., Liu B. Thermodynamically driven one-dimensional evolution of anatase TiO₂ nanorods: one-step hydrothermal synthesis for emerging intrinsic superiority of dimensionality. *J. Am. Chem. Soc.* **136**, 15310-15318 (2014).

14. Wu W. Q., Huang F., Chen D., Cheng Y. B., Caruso R. A. Thin Films of Dendritic Anatase Titania Nanowires Enable Effective Hole-Blocking and Efficient Light-Harvesting for High-Performance Mesoscopic Perovskite Solar Cells. *Adv. Funct. Mater.* **25**, 3264-3272 (2015).
15. Resasco J., Dasgupta N. P., Rosell J. R., Guo J., Yang P. Uniform doping of metal oxide nanowires using solid state diffusion. *J. Am. Chem. Soc.* **136**, 10521-10526 (2014).
16. Wang C.-C., Hsueh Y.-C., Su C.-Y., Kei C.-C., Perng T.-P. Deposition of uniform Pt nanoparticles with controllable size on TiO₂-based nanowires by atomic layer deposition and their photocatalytic properties. *Nanotechnology* **26**, 254002 (2015).
17. Su Z., Wang L., Grigorescu S., Lee K., Schmuki P. Hydrothermal growth of highly oriented single crystalline Ta₂O₅ nanorod arrays and their conversion to Ta₃N₅ for efficient solar driven water splitting. *Chem. Commun.* **50**, 15561-15564 (2014).
18. Zheng F., Lu H., Guo M., Zhang M., Zhen Q. Hydrothermal preparation of WO₃ nanorod array and ZnO nanosheet array composite structures on FTO substrates with enhanced photocatalytic properties. *Journal of Materials Chemistry C* **3**, 7612-7620 (2015).
19. Zheng F., *et al.* Hydrothermal preparation, growth mechanism and supercapacitive properties of WO₃ nanorod arrays grown directly on a Cu substrate. *CrystEngComm* **18**, 3891-3904 (2016).
20. He J., *et al.* Hydrothermal growth and optical properties of Nb₂O₅ nanorod arrays. *Journal of Materials Chemistry C* **2**, 8185-8190 (2014).
21. Kong D., *et al.* Seed-assisted growth of α -Fe₂O₃ nanorod arrays on reduced graphene oxide: a superior anode for high-performance Li-ion and Na-ion batteries. *J. Mater. Chem. A* **4**, 11800-11811 (2016).
22. Cao L., *et al.* Vertically aligned cobalt oxide nanowires on graphene networks for high-performance lithium storage. *Nanotechnology* **25**, 445704 (2014).
23. Yu H. K., Lee J.-L. Growth mechanism of metal-oxide nanowires synthesized by electron beam evaporation: A self-catalytic vapor-liquid-solid process. *Sci. Rep.* **4**, 6589 (2014).
24. Shen Y., *et al.* Epitaxy-Enabled Vapor-Liquid-Solid Growth of Tin-Doped Indium Oxide Nanowires with Controlled Orientations. *Nano Lett.* **14**, 4342-4351 (2014).
25. Sallet V., Sartel C., Vilar C., Lusson A., Galtier P. Opposite crystal polarities observed in spontaneous and vapour-liquid-solid grown ZnO nanowires. *Appl. Phys. Lett.* **102**, 182103 (2013).
26. Cheng G., *et al.* Large anelasticity and associated energy dissipation in single-crystalline nanowires. *Nat. Nanotech.* **10**, 687-691 (2015).
27. Klamchuen A., *et al.* Rational Concept for Designing Vapor-Liquid-Solid Growth of Single Crystalline Metal Oxide Nanowires. *Nano Lett.* **15**, 6406-6412 (2015).
28. Nagashima K., *et al.* Tailoring Nucleation at Two Interfaces Enables Single Crystalline NiO Nanowires via Vapor-Liquid-Solid Route. *ACS Appl. Mater. Interfaces* **8**, 27892-27899 (2016).
29. Li L., Zhang X., Li L., Zhai X., Zeng C. Magnetoresistance of single-crystalline La_{0.67}Sr_{0.33}

- MnO₃/MgO nanorod arrays. *Solid State Commun.* **171**, 46-49 (2013).
30. Domènech-Gil G., *et al.* Gas sensors based on individual indium oxide nanowire. *Sensors and Actuators B: Chemical* **238**, 447-454 (2017).
 31. Xu L., *et al.* Catalyst-free, selective growth of ZnO nanowires on SiO₂ by chemical vapor deposition for transfer-free fabrication of UV photodetectors. *ACS Appl. Mater. Interfaces* **7**, 20264-20271 (2015).
 32. Chen C., *et al.* Growth and characterization of well-aligned densely-packed rutile TiO₂ nanocrystals on sapphire substrates via metal-organic chemical vapor deposition. *Nanotechnology* **19**, 075611 (2008).
 33. Annanouch F. E., *et al.* Aerosol-assisted CVD-grown WO₃ nanoneedles decorated with copper oxide nanoparticles for the selective and humidity-resilient detection of H₂S. *ACS Appl. Mater. Interfaces* **7**, 6842-6851 (2015).
 34. Annanouch F. E., *et al.* Aerosol-assisted CVD-grown PdO nanoparticle-decorated tungsten oxide nanoneedles extremely sensitive and selective to hydrogen. *ACS Appl. Mater. Interfaces* **8**, 10413-10421 (2016).
 35. Lugo-Ruelas M., *et al.* Synthesis, microstructural characterization and optical properties of CuO nanorods and nanowires obtained by aerosol assisted CVD. *Journal of Alloys and Compounds* **643**, S46-S50 (2015).
 36. Deng K., Lu H., Shi Z., Liu Q., Li L. Flexible Three-Dimensional SnO₂ Nanowire Arrays: Atomic Layer Deposition-Assisted Synthesis, Excellent Photodetectors, and Field Emitters. *ACS Appl. Mater. Interfaces* **5**, 7845-7851 (2013).
 37. Hosein I. D., Hegde M., Jones P. D., Chirmanov V., Radovanovic P. V. Evolution of the faceting, morphology and aspect ratio of gallium oxide nanowires grown by vapor-solid deposition. *J. Cryst. Growth* **396**, 24-32 (2014).
 38. Zhao C., *et al.* Large-scale synthesis of bicrystalline ZnO nanowire arrays by thermal oxidation of zinc film: growth mechanism and high-performance field emission. *Cryst. Growth Des.* **13**, 2897-2905 (2013).
 39. Zhang Q., *et al.* Facile large-scale synthesis of vertically aligned CuO nanowires on nickel foam: Growth mechanism and remarkable electrochemical performance. *J. Mater. Chem. A* **2**, 3865-3874 (2014).
 40. Rackauskas S., *et al.* In situ study of noncatalytic metal oxide nanowire growth. *Nano Lett.* **14**, 5810-5813 (2014).
 41. Wang J., *et al.* Three-dimensional hierarchical Co₃O₄/CuO nanowire heterostructure arrays on nickel foam for high-performance lithium ion batteries. *Nano Energy* **6**, 19-26 (2014).
 42. Li A., Song H., Zhou J., Chen X., Liu S. CuO nanowire growth on Cu₂O by in situ thermal oxidation in air. *CrystEngComm* **15**, 8559-8564 (2013).

43. Tang C., *et al.* Enhanced adhesion and field emission of CuO nanowires synthesized by simply modified thermal oxidation technique. *Nanotechnology* **27**, 395605 (2016).
44. Kargar A., *et al.* ZnO/CuO heterojunction branched nanowires for photoelectrochemical hydrogen generation. *ACS nano* **7**, 11112-11120 (2013).
45. Zhang Z., *et al.* Atomic-Scale Observation of Vapor–Solid Nanowire Growth via Oscillatory Mass Transport. *ACS nano* **10**, 763-769 (2015).

3. How can one verify experimentally the strain-relaxation driven growth mechanism as proposed in Figure 7?

(A) First of all, it need to know what the driving force is to form AgCl NRs. And then we will explain the strain-relaxation driven growth mechanism In Fig 7.

(1) Formation of NRs on the flexible substrate.

The driving force for the growth of NRs could be the strain energy, originating from a lattice mismatch between the Ag film and AgCl nanorod. The Cl₂ plasma treatment allows the Ag film to react with Cl radicals, and to form an AgCl layer. The AgCl is subjected to compressive strain because of the lattice mismatch between Ag (lattice constant $a_{\text{Ag}} = 4.709\text{\AA}$) and AgCl ($a_{\text{AgCl}} = 5.545\text{\AA}$). To relieve the compressive strain, AgCl nanorods began to vertically grow on the surface of Ag film. The AgCl NRs could be epitaxially grown with the growth direction that minimize the lattice mismatch. It was experimentally found that the Ag film has (111) orientation, and the growth direction of AgCl NRs was confirmed to be (200) according to HR-TEM and XRD results. Since the lattice mismatch between Ag (111) and AgCl (200) is the smallest among all possible planes of Ag and AgCl, we concluded that the AgCl NRs are grown via the strain relaxation process.

(2) Growth mechanism of NRs on the flexible substrate.

The growth rate of NRs dramatically changes, with the type of substrate, as shown in Fig A3. The AgCl NRs grow rapidly on the flexible PI film in comparison with the glass substrate. It is notable that the PI film flexed convexly (Fig. A4a) during the NRs growth. This could be due to the strain induced by the lattice mismatch between AgCl NRs and Ag film. In order to experimentally confirm the reason for the rapid growth on PI, we attached the Ag-coated PI film onto a rigid substrate, the sample chuck in the plasma chamber, using a double-sided kapton tape, not to be bending by the lattice-mismatched strain, and then it was exposed to Cl₂ plasma (Fig. A4b). The plasma time is set to be 45 s. It was found that no NRs were grown on the fixed PI film, as in the case of the growth of NRs on the glass substrate (Fig. A4c). Finite element method (FEM) was employed to understand the effect of the substrate bending on the growth of NRs. As the PI film flex convexly, the strain gradient was localized at the

edge of NRs (Fig. A5). The strain gradient at the edge of NRs was calculated as a function of bending radius. The strain gradient enhanced with the decrease of bending radius, namely the increase of substrate flexibility (Fig. A6). Since the strain gradient activates the strain-induced diffusion of Ag atoms, the growth of AgCl NRs could be accelerated. The localization of the strain gradient can increase the strain-induced diffusion.

$$J = -D\nabla C - \frac{C\Omega DE}{k_B T} \nabla \varepsilon \approx -\frac{C\Omega DE}{k_B T} \nabla \varepsilon,$$

where J is the atomic flux, C , the atomic concentration, Ω , the atomic volume, D , the local diffusion coefficient, E , the Young's modulus, k_B , Boltzmann's constant, T , absolute temperature, and ε is the strain. In general, strain-induced diffusion is more dominant than that induced by the concentration gradient at low temperatures. Based on the experiment, we conclude that the substrate bending critically contributes to the rapid growth of NRs, due to strain-induced diffusion.

Various kinds of substrate including PI film, PET film, PC film, glass and sapphire were examined using the FEM analysis. The elastic property of those substrates are summarized in Table A2. The bending radius (r) of those substrates was calculated by the Stoney equation.

$$r = \frac{E_s t_s^2}{6(1 - \nu_s) \sigma_f t_f}$$

where r is the radius of curvature, σ_f , the film stress, E_s , the Young's modulus of the substrate, ν_s , its Poisson's ratio, t_s , the substrate thickness, and t_f is the film thickness. The calculated r values of substrates were normalized to the PI's one. The r values were determined to be $r \equiv 1$ for PI, $r = 1.69$ for PET, $r = 25$ for PC, $r = 1243$ for glass and $r = 2957$ for sapphire (Table A2). The strain gradient was plotted as a function of the relative bending radius (Fig. A6). As the degree of convex flexion increased, the strain gradient was more localized at the edge of the NRs

Figure A3. Time evolution of Ag film during Cl₂ plasma treatment on different thick substrates, (a) 100-μm-thick polyimide (PI), (b) 130-μm-thick polyethylene terephthalate (PET), and (c) 700-μm-thick soda lime glass. Scale bar: 1 μm.

Figure A4. 300-nm-thick Ag coated on (a) PI film, (b) inflexible PI film fixed by tape and (c) glass were exposed Cl₂ plasma for 45 s. (Left) the photographs of the substrates during Cl₂ plasma exposure. (Right) the SEM images after plasma treatment

Figure A5. Calculated gradient of strain under volume expansion from Ag to AgCl on (e) rigid glass and (f) flexible PI film. Volume expansion was assumed to occur at interface of AgCl nanorods with Ag film. In flexible PI film, strain was localized at bottom of AgCl nanorods.

Table A2. Elastic property and thickness of substrates

Substrates	Young's modulus [GPa]	Possion's ratio	Substrate thickness [μm]	Relative bending radius
PI	2.5	0.4	100	1
PET	2.5	0.4	130	1.69
PC	2.5	0.4	500	25
Glass	74	0.3	700	1243
Sapphire	345	0.3	500	2957

Figure A6. Strain gradient at edge of nanorods was calculated as a function of relative bending radius. Relative bending radius was normalized bending radius to that of PI film. PI, PET, PC film, glass, and sapphire were noted.

We revised the manuscript in accordance with the reviewer's comments on experimental verification of the strain-relaxation driven growth mechanism in Figure 7.

We reorganize the sequence of Figures 6 and 7 in the revised manuscript in order to clearly explain the strain-relaxation driven growth mechanism.

The Figure 6 is revised in the manuscript

Figure 6. Effect of substrate flexibility on growth of AgCl nanorods. XRD patterns of Cl_2 -exposed Ag film as a function of exposure time on (a) polyimide and (b) soda-lime glass; Ag (black squares); AgCl (red circles) (c) Partial AgCl ratio calculated from XRD patterns. (d) Photography of Ag coated 100- μm -thick, 130- μm -thick PET, 500- μm -thick PC films, and 700- μm -thick soda lime glass during Cl_2 exposure

Figure 6. Schematic illustration of strain-relaxation driven growth mechanism of AgCl nanorods.

(a) Ag coated PI film was exposed to Cl₂ plasma. (b) In early stage of growth, Ag surface reacted with Cl radicals, producing AgCl islands due to volume expansion from Ag to AgCl. PI film bent convexly in response of strain. (c) Cl radicals diffused into Ag/AgCl interfaces; as a result, compressive strain was applied to AgCl and acted as driving force of nanorod growth. Interfacial lattice mismatch (Ag/AgCl) guided (200)-preferred grow of AgCl nanorod. (d) AgCl nanorods grow on PI film as a result of strain relaxation. (e) Atomic configuration of Ag/AgCl interface. Compressive strain in the AgCl stimulates growth of nanorods. Base of nanorod was subjected to less strain due to strain relaxation of nanorods. (f) During growth of nanorod, Ag atoms diffuse to its base because of strain gradient

The Figure 7 is revised in the manuscript

Figure.7 Schematic illustration of strain-relaxation driven growth mechanism of AgCl nanorods.

Calculated gradient of strain under volume expansion from Ag to AgCl on (a) rigid glass and (b) flexible PI film. Volume expansion was assumed to occur at interface of AgCl nanorods with Ag film. In flexible PI film, strain was localized at bottom of AgCl nanorods. (c) Ag coated PI film was exposed to Cl_2 plasma. (d) At early stage of growth, Ag surface reacted with Cl radicals, producing AgCl islands due to volume expansion from Ag to AgCl. PI film bent convexly in response of strain. (e) Cl radicals diffused into Ag/AgCl interfaces; as a result, compressive strain was applied to AgCl and acted as driving force of nanorod growth. Interfacial lattice mismatch (Ag/AgCl) guided (200)-preferred grow of AgCl nanorod. (f) AgCl nanorods grow on PI film as a result of strain relaxation. (g) Atomic configuration of Ag/AgCl interface. Compressive strain in AgCl stimulates growth of the nanorods. Base of nanorod was subjected to less strain due to strain relaxation of nanorods. (h) during growth, Ag atoms diffuse to the base of nanorods because of strain gradient

Figure 7. Effect of substrate flexibility on growth of AgCl nanorods. XRD patterns of Cl_2 -exposed Ag film as a function of exposure time on (a) polyimide and (b) soda-lime glass; Ag (black squares); AgCl (red circles) (c) Partial AgCl ratio calculated from XRD patterns. (d) Photography of Ag coated 100- μm -thick, 130- μm -thick PET, 500- μm -thick PC films, and 700- μm -thick soda lime glass during Cl_2 exposure. Calculated gradient of strain under volume expansion from Ag to AgCl on (e) rigid glass and (f) flexible PI film. Volume expansion was assumed to occur at interface of AgCl nanorods with Ag film. In flexible PI film, strain was localized at bottoms of AgCl nanorods.

We rewrote a paragraph in the revised manuscript (page 9, line 20 ~ page 10, line 7).

On the basis of these results, we proposed a growth mechanism of the AgCl NRs (Fig. 7). When Ag was deposited, the face-centered-cubic Ag film tended to grow with (111) preferred orientation (Fig. 5d). At the initial stage of growth, Ag atoms on the Ag surface reacted with Cl radicals from the Cl_2 plasma (Fig. 7c). When Ag is converted to AgCl, large lattice mismatch between Ag (lattice constant $a = 4.079\text{\AA}$) and AgCl ($a = 5.545\text{\AA}$) leads to strain in AgCl nanodots (Fig. 7d).⁵⁰ To minimize strain energy, the nucleated AgCl nanodots had (200) orientation, having the smallest lattice mismatch with Ag (111) film. The AgCl nanodots acted as a seed for the growth of AgCl NRs. The AgCl NRs grew from the base, where the diffused Ag atoms were chlorized by absorbed Cl radicals at the interface of the NRs with Ag film. During the reaction, the AgCl underwent compressive strain from the remaining

Ag film due to its smaller lattice constant (Fig. 7g). To relieve this strain, the AgCl NRs grew on the surface of AgCl.

→ **Driving force of growth of AgCl NRs.** On the basis of these results, we proposed a growth mechanism of the AgCl NRs (Fig. 6). When Ag was deposited, the face-centered-cubic Ag film tended to grow with the preferred orientation of (111) (Fig. 5d). During the initial stage of growth, Ag atoms on the surface of Ag film reacted with Cl radicals from the Cl₂ plasma (Fig. 6a). Conversion of Ag to AgCl nanodots imposes a large strain because of the lattice mismatch between Ag (lattice constant $a_{\text{Ag}} = 4.079\text{\AA}$) and AgCl ($a_{\text{AgCl}} = 5.545\text{\AA}$) (Fig. 6b). To minimize the strain energy, the nucleated AgCl nanodots had (200) orientation, which has the smallest lattice mismatch with Ag (111) film. Because the AgCl nanodots acted as a seed for the growth of AgCl NRs, the Ag atoms were reacted with adsorbed Cl radicals at the interface of the NRs with Ag film (Fig. 6c). At that time, the AgCl underwent a compressive strain from the remaining Ag (Fig. 6e); to relieve this strain, NRs grew vertically.

We revised a paragraph in the revised manuscript (page 7, line 23 ~ page 8, line 11)

Growth mechanism of the AgCl NRs. As shown in Fig. 2, the AgCl NRs grew rapidly as substrate became thin. To investigate the dependence of growth characteristics on the kind of substrate, XRD was measured on Ag-deposited PI film (Fig. 6a), glass (Fig. 6b), and c-sapphire (Supplementary Fig. S6) as a function of plasma time. As-deposited Ag film showed only metallic Ag peaks. As the plasma time increased, AgCl peaks began to appear and the Ag peaks weakened. The AgCl ratio (Fig. 6c) (i.e., [area of AgCl peak]/[total areas of AgCl and Ag peaks]) reached 1 within 20 s on the PI film but not until 120 s on the glass and on the c-sapphire; i.e., AgCl grew six times faster on PI film than on glass and growth rate. This fast growth on PI is due to its spontaneously convex shape during the Cl₂ plasma treatment (Fig. 6d). Under the same plasma condition, flexible thin PI and PET film bent, whereas inflexible thick PC film and glass did not bend. During NR growth, the larger lattice constant of AgCl than that of Ag induced volume expansion. At the interface of Ag with AgCl, the lattice mismatch imposes compressive stress on AgCl, so the substrate bends to minimize strain energy.

→ **The growth mechanism of AgCl NRs on flexible film.** The AgCl NRs grew rapidly as the substrate became thin (Fig. 2). To investigate the dependence of growth characteristics on the kind of substrate, XRD was measured on Ag-deposited PI film (Fig. 7a), glass (Fig. 7b), and c-sapphire (Supplementary Fig. S6) as a function of plasma time. As-deposited Ag film showed only metallic Ag peaks. As the plasma time increased, AgCl peaks began to appear and the Ag peaks weakened. The ratio of AgCl (Fig. 7c) (i.e., [area of AgCl peak]/[total areas of AgCl and Ag peaks]) reached 1.0 within 20 s on the PI film but it took 120 s on both the glass and on the c-sapphire; i.e., AgCl grew six times faster on PI film than on the glass and on the c-sapphire. This fast growth on PI is due to the spontaneous bending with convex shape during the Cl₂ plasma treatment (Fig. 7d).

The paragraph corresponding to Table S1 was moved to supplementary information. (page 8, line 11 ~ page 8, line 22)

The relationship between radius (r) of curvature of the substrate and film stress (σ_f) is given by the Stoney equation

$$r = \frac{E_s t_s^2}{6(1 - \nu_s) \sigma_f t_f}$$

where E_s is the Young's modulus of the substrate and ν_s is its Poisson's ratio, t_s is the substrate thickness, and t_f is the film thickness. Calculated r were normalized to that of PI; they were: PI, $r \equiv 1$; PET, $r = 1.69$; PC, $r = 25$; glass $r = 1243$ (Supplementary Table S1). Because the polymer films (PI, PET, and PC) had similar elastic properties ($E_s = 2.5$ GPa, $\nu_s = 0.4$), their radius of curvature was only proportional to square of film thickness. Thus, the radius of curvature was the smallest on 100- μm -thick PI film, and increased in the sequence of 130- μm -thick PET and 500- μm -thick PC film (Fig. 6d). The 700- μm -thick glass had larger Young's modulus of 74 GPa than the polymers, so its r was expected to 1,000 times larger than that of PI film, so flexion of the glass was almost absent during volume expansion.

The paragraph was added in the revised manuscript (page 8, line 11)

→ This difference occurs because the PI film has the smallest bending radius due to its low thickness (d) and low Young's modulus (E) from the Stoney equation⁴⁷ (PI film: $d_{\text{PI}} = 100$ μm , $E_{\text{PI}} = 2.5$ GPa; glass: $d_{\text{glass}} = 700$ μm , $E_{\text{glass}} = 74$ GPa). The details are described in supplementary Table S1.

The paragraph was revised in the revised manuscript (page 8, line 23 ~ page 9, line 19)

To investigate the effect of substrate bending on the growth, we conducted finite element method (FEM) analysis on rigid glass (Fig. 7a) and flexible PI film (Fig. 7b). The reaction of Ag with Cl was assumed to expand in volume at the interface of NRs with Ag film. During the expansion, the soft PI film ($E = 2.5$ GPa) bent convexly, whereas glass was not bent due to its high stiffness ($E = 74$ GPa). Flexion of the PI substrate expanded the distance between adjacent AgCl NRs and allowed easy movement of Cl to the remaining Ag. The flexion also caused strain distribution to vary gradually at the AgCl layer (Supplementary Fig. 7b) because the bending relaxes the strain. As a result, the strain gradient of 5.72 μm^{-1} was localized near the edge of the nanorods in the PI film (Fig. 7b). However, in the case of glass, each layer underwent uniform strain (Supplementary Fig. 7a) because high stiffness of the glass constrained the in-plane expansion, so the strain gradient on the glass was only of 0.24 μm^{-1} (Fig. 7a). The localization of the strain gradient can increase the strain-induced diffusion of Ag atoms as^{47,48}

$$J = -D\nabla C - \frac{C\Omega DE}{k_B T} \nabla \varepsilon \approx - \frac{C\Omega DE}{k_B T} \nabla \varepsilon,$$

where J is the atomic flux, C is the atomic concentration, Ω is the atomic volume, D is the local diffusion coefficient, E is the Young's modulus, k_B is Boltzmann's constant, T is absolute temperature, and ε is the strain. In general, strain-induced diffusion is much larger than concentration gradient diffusion at low temperature,⁴⁹ so the concentration gradient was neglected (Supplementary Table S2). Because the strain gradient was localized at the edge of the NRs on the PI film, diffusion of Ag atoms into the edge of the NRs was facilitated, so NRs growth was accelerated on the PI film. (Fig. 6c)

→ To further investigate the effect of substrate bending on the growth of NRs, we conducted finite element method (FEM) analysis on rigid glass (Fig. 7e) and flexible PI film (Fig. 7f). The reaction of Ag with Cl causes to expand the volume of Ag layer because $a_{AgCl} = 5.545 \text{ \AA} > a_{Ag} = 4.079 \text{ \AA}$. This causes convex bending of PI film, but not of the stiff glass. The bending relaxes the strain, so flexion of the PI film caused strain distribution to vary rapidly at the Ag/AgCl interface (Supplementary Fig. 7b) because the bending relaxes the strain. As a result, the strain gradient of $11.0 \mu\text{m}^{-1}$ was localized near the edge of the NRs in the PI film (Fig. 7f). However, in the case of glass, there was no strain gradient (Supplementary Fig. 7a) because of the high stiffness of the glass. The strain gradient on the glass was calculated to be only $0.79 \mu\text{m}^{-1}$ (Fig. 7e). The localization of strain gradient can increase the strain-induced diffusion of Ag atoms as^{47,48}

$$J = -D\nabla C - \frac{C\Omega DE}{k_B T} \nabla \varepsilon \approx - \frac{C\Omega DE}{k_B T} \nabla \varepsilon,$$

where J is the atomic flux, C , the atomic concentration, Ω , the atomic volume, D , the local diffusion coefficient, E , the Young's modulus, k_B , Boltzmann's constant, T , absolute temperature, and ε is the strain. In general, strain-induced diffusion is more dominant than that induced by the concentration gradient at low temperatures,⁴⁹ so the concentration gradient was neglected (Supplementary Table S2).

The supplementary figure was added in the revised manuscript.

Supplementary Figure S8. Effect of substrate flexibility on strain gradient. Strain gradient at edge of nanorods was calculated as a function of relative bending radius. Relative bending radius was normalized bending radius to that of PI film. PI, PET, glass, and sapphire were noted.

The supplementary Table S1 was revised in the revised manuscript.

Substrates	Young's modulus (E _s) [GPa]	Possion's ratio (ν _s)	Substrate thickness (t _s) [μm]	Relative radius of curvature
PI	2.5	0.4	100	1
PET	2.5	0.4	130	1.69
PC	2.5	0.4	500	25
Glass	74	0.3	700	1243.2

Supplementary Table S1. Elastic properties and thickness of the substrates. Relative radius of curvature was calculated using Stoney equation, then normalized to that of PI film. Stress-thickness products of film ($\sigma_f t_f$) were assumed to be constant on each substrate.

→

Substrates	Young's modulus (E_s) [GPa]	Poisson's ratio (ν_s)	Substrate thickness (t_s) [μm]	Relative radius of curvature
PI	2.5	0.4	100	1
PET	2.5	0.4	130	1.69
PC	2.5	0.4	500	25
Glass	74	0.3	700	1243.2
Sapphire	345	0.3	500	2957

Supplementary Table S1. Elastic properties and thickness of substrates. Relative radius of curvature was calculated using Stoney equation, then normalized to that of PI film. Stress-thickness products of film ($\sigma_f t_f$) were assumed to be constant on each substrate.

The sentences was added in the revised manuscript (page 9, line 10)

→ The strain gradient was calculated on the various substrates such as PI, PET, PC film, glass, and sapphire. The elastic properties of substrates were summarized in Supplementary Table S1. The bending radius of substrates were calculated using the Stoney equation⁴⁷ and normalized to that of PI (Supplementary Table S1). The strain gradient at the edge of NRs was plotted as a function of relative bending radius (Supplementary Fig. S8). As the degree of convex flexion increased, the strain gradient became increasingly localized at the edge of the NRs.

The sentence was revised in the revised manuscript (page 9, line 17)

Because the strain gradient was localized at the edge of the NRs on the PI film, diffusion of Ag atoms into the edge of the NRs was facilitated, so NRs growth was accelerated on the PI film. (Fig. 6c)

→ Because the strain gradient was localized at the edge of the NRs on the PI film, strain-induced diffusion drove facilitated diffusion of Ag atoms toward the bases of the NRs, so NRs growth was accelerated on the PI film. Experimental results provide the evidence that Ag atoms diffuse to the bases of the NRs and react with Cl radicals until Ag is completely consumed, leading to the formation of single-crystal AgCl NRs.

The supplementary Figure S8 was replaced with Figure 8.

Supplementary Figure S8. Critical role of substrate bendability to produce AgCl nanorods. SEM images of Cl₂ plasma-exposed Ag on PI film attached to chuck by kapton tape.

→

Figure 8. Effect of bending of substrate on growth of nanorods. 300-nm-thick Ag coated on (a) PI film, (b) inflexible PI film fixed by tape and (c) glass were exposed Cl₂ plasma for 45 s. (Left) Photographs of the substrates during Cl₂ plasma exposure. (Right) SEM images after plasma treatment

The sentence was added in the revised manuscript (page 10, line 19)

→ To test this hypothesized mechanism of rapid growth on PI, we attached the Ag-coated PI film onto a rigid substrate (i.e., the sample chuck in the plasma chamber) using double-sided kapton tape, so that the substrate could not bend under lattice-mismatch strain. And then it was exposed to Cl₂ plasma (Fig. 8). The plasma time is set to be 45 s. It was found that no NRs were grown on the fixed PI film, as in the case of the rigid glass substrate. Based on the experiment, we conclude that the substrate bending critically contributes to the rapid growth of NRs, due to strain-induced diffusion.

<For Reviewer #2>

The authors of this manuscript described the preparation of single-crystalline AgCl nanorods by using a Cl₂ plasma source. A growth mechanism based on a strain-relaxation process was proposed.

The manuscript is well written and the topic is of interest. However, I would recommend submission to a lower impact journal as the novelty of the work does not justify publication in Nature Communications –impact factor 11.329.

(A) In this work, we for the first time demonstrate the single-crystalline AgCl nanorods (NRs) by using a Cl₂ plasma source. Vertically well-aligned AgCl NRs were grown by a strain-relaxation process on flexible polymer substrates. This plasma-assisted NRs showed extremely high growth rate at low temperature compared with the previously reported nanorod growth techniques (Table A1). Furthermore, the growth rate of the AgCl NRs can be dramatically improved to ~2,000 nm/min on flexible polymer substrate. This is an immediate technology which is applicable to a large area substrate using a roll-to-roll process. No lithography using mask molds, and thermal processes are required in the proposed method. Thus, the AgCl nanorod must be a significant impact on highly efficient flexible optoelectronic devices.

On the basis of experimental results and theoretical calculations, we discussed the driving force and growth mechanism of AgCl NRs on the flexible film. The lattice constant of AgCl ($a_{\text{AgCl}} = 5.545\text{\AA}$) is larger than that of Ag ($a_{\text{Ag}} = 4.709\text{\AA}$), so the AgCl is subjected to compressive strain at its interface with Ag film. To minimize strain energy, the AgCl NRs began to vertically grow toward the surface normal direction, [200], because AgCl (200) has the smallest lattice mismatch with Ag (111). The epitaxial relationship between AgCl (200) and Ag (111) is matched to be cube-on-hexagon geometry. Strain induced by lattice mismatch causes the substrate to flex, and thereby induce a strain gradient at the periphery of AgCl NRs. The gradient causes a strain-induced diffusion of Ag atoms and accelerates growth of the NRs.

We also confirm that the size of AgCl NRs were tunable from sub-wavelength (< 400 nm) to wavelength scale (> 400 nm) by adjusting the thickness of Ag. Thus, optical behavior could be managed from near-zero haze (0.23%) to full-scattering (100 %). The organic light emitting diodes with the wavelength scale AgCl nanorods showed a remarkable enhancement in luminance efficiency (66.9 cd/A at 1,000 cd/m²) up to 33%, compared to that of control device (50.3 cd/A at 1,000 cd/m²). The wavelength-scale nanorods were very effective in extracting the confined wave-guided electromagnetic wave in the substrate.

Table A1. Growth temperature and rate of various inorganic nanorods

Growth Method	Materials	Growth Temperature [°C]	Growth Rate [nm/min]	Ref.
Hydrothermal	ZnO	90	0.6	1
	ZnO	60	1	2
	ZnO	95	6.25	3
	ZnO	70	2.9	4
	ZnO	79	11.4	5
	ZnO	90	5	6
	ZnO	90	6	7
	ZnO	100	2.8	8
	TiO ₂	140	3.7	9
	TiO ₂	150	1.2	10
	TiO ₂	90	17.1	11
	TiO ₂	150	5.6	12
	TiO ₂	170	0.23	13
	TiO ₂	190	23	14
	TiO ₂	210	10	15
	TiO ₂	200	2.08	16
	Ta ₂ O ₅	240	0.41	17
	WO ₃	170	7.5	18
	WO ₃	180	6.25	19
	Nb ₂ O ₅	150	4	20
	Fe ₂ O ₃	60	1.3	21
	CoO	120	17	22
Vapor-liquid-solid	ITO	300	30	23
	ITO	800	16.3	24
	ZnO	990	100	25
	ZnO	820	230	26
	ZnO	750	15	27
	NiO	550	30	28
	MgO	925	1.7	29
	SnO ₂	750	5	27
	In ₂ O ₃	750	15	27

	In ₂ O ₃	750	-	30
	MgO	750	17	27
Chemical Vapor Deposition	ZnO	650	90	31
	TiO ₂	550	-	32
	WO ₃	500	61	33
	WO ₃	500	108	34
	CuO	600	900	35
	SnO ₂	760	55.5	36
	Ga ₂ O ₃	1000	120	37
Thermal oxidation	ZnO	500	33.3	38
	CuO	400	13.8	39
	CuO	400	5.4	40
	CuO	400	14	41
	CuO	600	9.7	42
	CuO	400	11.1	43
	CuO	400	20	44
	W ₁₈ O ₄₉	700	12	45
AgCl	This work	~ RT	2000	

1. Cheng J. J., Nicaise S. M., Berggren K. K., Gradec ˇak S. Dimensional tailoring of hydrothermally grown zinc oxide nanowire arrays. *Nano Lett.* **16**, 753-759 (2015).
2. Joo J., Chow B. Y., Prakash M., Boyden E. S., Jacobson J. M. Face-selective electrostatic control of hydrothermal zinc oxide nanowire synthesis. *Nat. Mater.* **10**, 596-601 (2011).
3. Park G. C., *et al.* Hydrothermally grown In-doped ZnO nanorods on p-GaN films for color-tunable heterojunction light-emitting-diodes. *Sci. Rep.* **5**, 10410 (2015).
4. Lee J. M., No Y.-S., Kim S., Park H.-G., Park W. I. Strong interactive growth behaviours in solution-phase synthesis of three-dimensional metal oxide nanostructures. *Nat. commun.* **6**, 6325 (2015).
5. Watanabe K., *et al.* Arbitrary cross-section SEM-cathodoluminescence imaging of growth sectors and local carrier concentrations within micro-sampled semiconductor nanorods. *Nat. commun.* **7**, 10609 (2016).
6. Kim B. H., Kwon J. W. Metal catalyst for low-temperature growth of controlled zinc oxide nanowires on arbitrary substrates. *Sci. Rep.* **4**, 4379 (2014).
7. Consonni V., *et al.* Selective area growth of well-ordered ZnO nanowire arrays with controllable polarity. *ACS nano* **8**, 4761-4770 (2014).
8. Yue H. Y., *et al.* ZnO nanowire arrays on 3D hierarchical graphene foam: biomarker detection of

- Parkinson's disease. *ACS nano* **8**, 1639-1646 (2014).
9. Ye M., Liu H. Y., Lin C., Lin Z. Hierarchical Rutile TiO₂ Flower Cluster-Based High Efficiency Dye-Sensitized Solar Cells via Direct Hydrothermal Growth on Conducting Substrates. *Small* **9**, 312-321 (2013).
 10. Huang H., *et al.* Hydrothermal Growth of TiO₂ Nanorod Arrays and In Situ Conversion to Nanotube Arrays for Highly Efficient Quantum Dot-Sensitized Solar Cells. *Small* **9**, 3153-3160 (2013).
 11. Yang T., *et al.* Position-controlled hydrothermal growth of periodic individual ZnO nanorod arrays on indium tin oxide substrate. *J. Phys. Chem. C* **118**, 20613-20619 (2014).
 12. Berhe S. A., Nag S., Molinets Z., Youngblood W. J. Influence of Seeding and Bath Conditions in Hydrothermal Growth of Very Thin (~ 20 nm) Single-Crystalline Rutile TiO₂ Nanorod Films. *ACS Appl. Mater. Interfaces* **5**, 1181-1185 (2013).
 13. Chen J., Yang H. B., Miao J., Wang H.-Y., Liu B. Thermodynamically driven one-dimensional evolution of anatase TiO₂ nanorods: one-step hydrothermal synthesis for emerging intrinsic superiority of dimensionality. *J. Am. Chem. Soc.* **136**, 15310-15318 (2014).
 14. Wu W. Q., Huang F., Chen D., Cheng Y. B., Caruso R. A. Thin Films of Dendritic Anatase Titania Nanowires Enable Effective Hole-Blocking and Efficient Light-Harvesting for High-Performance Mesoscopic Perovskite Solar Cells. *Adv. Funct. Mater.* **25**, 3264-3272 (2015).
 15. Resasco J., Dasgupta N. P., Rosell J. R., Guo J., Yang P. Uniform doping of metal oxide nanowires using solid state diffusion. *J. Am. Chem. Soc.* **136**, 10521-10526 (2014).
 16. Wang C.-C., Hsueh Y.-C., Su C.-Y., Kei C.-C., Perng T.-P. Deposition of uniform Pt nanoparticles with controllable size on TiO₂-based nanowires by atomic layer deposition and their photocatalytic properties. *Nanotechnology* **26**, 254002 (2015).
 17. Su Z., Wang L., Grigorescu S., Lee K., Schmuki P. Hydrothermal growth of highly oriented single crystalline Ta₂O₅ nanorod arrays and their conversion to Ta₃N₅ for efficient solar driven water splitting. *Chem. Commun.* **50**, 15561-15564 (2014).
 18. Zheng F., Lu H., Guo M., Zhang M., Zhen Q. Hydrothermal preparation of WO₃ nanorod array and ZnO nanosheet array composite structures on FTO substrates with enhanced photocatalytic properties. *Journal of Materials Chemistry C* **3**, 7612-7620 (2015).
 19. Zheng F., *et al.* Hydrothermal preparation, growth mechanism and supercapacitive properties of WO₃ nanorod arrays grown directly on a Cu substrate. *CrystEngComm* **18**, 3891-3904 (2016).
 20. He J., *et al.* Hydrothermal growth and optical properties of Nb₂O₅ nanorod arrays. *Journal of Materials Chemistry C* **2**, 8185-8190 (2014).
 21. Kong D., *et al.* Seed-assisted growth of α -Fe₂O₃ nanorod arrays on reduced graphene oxide: a superior anode for high-performance Li-ion and Na-ion batteries. *J. Mater. Chem. A* **4**, 11800-11811 (2016).

22. Cao L., *et al.* Vertically aligned cobalt oxide nanowires on graphene networks for high-performance lithium storage. *Nanotechnology* **25**, 445704 (2014).
23. Yu H. K., Lee J.-L. Growth mechanism of metal-oxide nanowires synthesized by electron beam evaporation: A self-catalytic vapor-liquid-solid process. *Sci. Rep.* **4**, 6589 (2014).
24. Shen Y., *et al.* Epitaxy-Enabled Vapor–Liquid–Solid Growth of Tin-Doped Indium Oxide Nanowires with Controlled Orientations. *Nano Lett.* **14**, 4342-4351 (2014).
25. Sallet V., Sartel C., Vilar C., Lusson A., Galtier P. Opposite crystal polarities observed in spontaneous and vapour-liquid-solid grown ZnO nanowires. *Appl. Phys. Lett.* **102**, 182103 (2013).
26. Cheng G., *et al.* Large anelasticity and associated energy dissipation in single-crystalline nanowires. *Nat. Nanotech.* **10**, 687-691 (2015).
27. Klamchuen A., *et al.* Rational Concept for Designing Vapor–Liquid–Solid Growth of Single Crystalline Metal Oxide Nanowires. *Nano Lett.* **15**, 6406-6412 (2015).
28. Nagashima K., *et al.* Tailoring Nucleation at Two Interfaces Enables Single Crystalline NiO Nanowires via Vapor–Liquid–Solid Route. *ACS Appl. Mater. Interfaces* **8**, 27892-27899 (2016).
29. Li L., Zhang X., Li L., Zhai X., Zeng C. Magnetoresistance of single-crystalline La 0.67 Sr 0.33 MnO 3/MgO nanorod arrays. *Solid State Commun.* **171**, 46-49 (2013).
30. Domènech-Gil G., *et al.* Gas sensors based on individual indium oxide nanowire. *Sensors and Actuators B: Chemical* **238**, 447-454 (2017).
31. Xu L., *et al.* Catalyst-free, selective growth of ZnO nanowires on SiO₂ by chemical vapor deposition for transfer-free fabrication of UV photodetectors. *ACS Appl. Mater. Interfaces* **7**, 20264-20271 (2015).
32. Chen C., *et al.* Growth and characterization of well-aligned densely-packed rutile TiO₂ nanocrystals on sapphire substrates via metal–organic chemical vapor deposition. *Nanotechnology* **19**, 075611 (2008).
33. Annanouch F. E., *et al.* Aerosol-assisted CVD-grown WO₃ nanoneedles decorated with copper oxide nanoparticles for the selective and humidity-resilient detection of H₂S. *ACS Appl. Mater. Interfaces* **7**, 6842-6851 (2015).
34. Annanouch F. E., *et al.* Aerosol-assisted CVD-grown PdO nanoparticle-decorated tungsten oxide nanoneedles extremely sensitive and selective to hydrogen. *ACS Appl. Mater. Interfaces* **8**, 10413-10421 (2016).
35. Lugo-Ruelas M., *et al.* Synthesis, microstructural characterization and optical properties of CuO nanorods and nanowires obtained by aerosol assisted CVD. *Journal of Alloys and Compounds* **643**, S46-S50 (2015).
36. Deng K., Lu H., Shi Z., Liu Q., Li L. Flexible Three-Dimensional SnO₂ Nanowire Arrays: Atomic Layer Deposition-Assisted Synthesis, Excellent Photodetectors, and Field Emitters. *ACS Appl. Mater. Interfaces* **5**, 7845-7851 (2013).

37. Hosein I. D., Hegde M., Jones P. D., Chirmanov V., Radovanovic P. V. Evolution of the faceting, morphology and aspect ratio of gallium oxide nanowires grown by vapor–solid deposition. *J. Cryst. Growth* **396**, 24-32 (2014).
38. Zhao C., *et al.* Large-scale synthesis of bicrystalline ZnO nanowire arrays by thermal oxidation of zinc film: growth mechanism and high-performance field emission. *Cryst. Growth Des.* **13**, 2897-2905 (2013).
39. Zhang Q., *et al.* Facile large-scale synthesis of vertically aligned CuO nanowires on nickel foam: Growth mechanism and remarkable electrochemical performance. *J. Mater. Chem. A* **2**, 3865-3874 (2014).
40. Rackauskas S., *et al.* In situ study of noncatalytic metal oxide nanowire growth. *Nano Lett.* **14**, 5810-5813 (2014).
41. Wang J., *et al.* Three-dimensional hierarchical Co₃O₄/CuO nanowire heterostructure arrays on nickel foam for high-performance lithium ion batteries. *Nano Energy* **6**, 19-26 (2014).
42. Li A., Song H., Zhou J., Chen X., Liu S. CuO nanowire growth on Cu₂O by in situ thermal oxidation in air. *CrystEngComm* **15**, 8559-8564 (2013).
43. Tang C., *et al.* Enhanced adhesion and field emission of CuO nanowires synthesized by simply modified thermal oxidation technique. *Nanotechnology* **27**, 395605 (2016).
44. Kargar A., *et al.* ZnO/CuO heterojunction branched nanowires for photoelectrochemical hydrogen generation. *ACS nano* **7**, 11112-11120 (2013).
45. Zhang Z., *et al.* Atomic-Scale Observation of Vapor–Solid Nanowire Growth via Oscillatory Mass Transport. *ACS nano* **10**, 763-769 (2015).

<For Reviewer #3>

1. The data is generally clearly presented and easy to interpret however some of the claims are extraordinarily inaccurate - it is unclear whether this is a fault of the authors or an issue with the use of English. For example on more than one occasion the authors claim deposition was achieved on 'any kind of substrate' (P3L12) and again "were grown on any substrate" (P3L22) - clearly only a few transparent (excluding sapphire for growth modelin) substates were used so i suggest this is canned to something like " a range transparent substrates were used, including rigid (glass) and flexible (Pet...)".

(A) As the reviewer's comment, we revised the manuscript as in below.

(Page 3, line 12)

Vertically well-aligned AgCl NRs were grown by a strain-relaxation process on any kind of substrate.

→ Vertically well-aligned AgCl NRs were grown by a strain-relaxation process on several kind of transparent substrate.

(Page 3, line 22)

Single-crystalline AgCl NRs were grown on any substrate with deposited Ag layer by introducing a chlorine plasma source.

→ Single-crystalline AgCl NRs were grown on transparent substrate with deposited Ag layer by introducing a chlorine plasma source.

2. The poor use of English detracts from the communication in the manuscript e.g. "Cl radicals react consistently"(abstract), nanostructure vs nanostructure (entire manuscript), "flexible plastic substrates are flexible..."(P2L1), "higher power efficiency than did a device" (P2last line)...however quality improves as the documents extends. One recurring issue is the use of "chlorized", which i assume is from silanized? this is improper and should be replaced with "completely reacted" or similar.

(A) As the reviewer's comment, we revised the manuscript as in below.

(Page 1, line 12)

We report a way to fabricate single-crystal AgCl nanorods (NRs) using a Cl₂ plasma source on Ag-coated substrate. Cl radicals react consistently with Ag to form AgCl NRs by a strain-relaxation process.

→ We report a way to fabricate single-crystal AgCl nanorods (NRs) using a Cl₂ plasma source on Ag-coated substrate. Cl radicals react with Ag to form AgCl NRs by a strain-relaxation process.

(Page 2, line 1)

Flexible plastic substrates are lightweight, inexpensive, flexible, and enable roll-to-roll mass production,¹⁻³ so they have applications as components in next-generation optoelectronic devices such as organic light-emitting diodes (OLEDs), displays, organic solar cells and photodiodes.

→ Flexible plastic substrates are lightweight, inexpensive, and enable roll-to-roll mass production,¹⁻³ so they have applications as components in next-generation optoelectronic devices such as organic light-emitting diodes (OLEDs), displays, organic solar cells and photodiodes.

(Page2, line 24)

because this structure minimized the waveguide mode in the substrate, OLEDs produced using this method had 84% higher power efficiency than did a device that used planar film.

→ because this structure minimized the waveguide mode in the substrate, the power efficiency of OLEDs was 84% higher than that of the device fabricated on planar film.

(Page6, line 11)

These results suggest that all of Ag was completely chlorized to produce AgCl NRs by the treatment for 90 s.

→ These results suggest that all of the Ag reacted completely to produce AgCl NRs by the treatment for 90 s.

(Page 10, line 2)

The AgCl nanodots acted as a seed for the growth of AgCl NRs. The AgCl NRs grew from the base, where the diffused Ag atoms were chlorized by absorbed Cl radicals at the interface of the NRs with Ag film.

→ Because the AgCl nanodots acted as a seed for the growth of AgCl NRs, the Ag atoms were reacted with adsorbed Cl radicals at the interface of the NRs with Ag film.

(Figure 1, Caption)

During the plasma treatment, Ag was completely chlorized to produce AgCl nanorods on flexible plastic substrates.

→ During the plasma treatment, Ag reacted completely to produce AgCl nanorods on flexible plastic substrates.

(Figure S1, caption)

After Cl₂ plasma treatment on Ag, all of Ag was completely chlorized to produce AgCl compounds.

→ After Cl₂ plasma treatment, all Ag had reacted completely to produce AgCl compounds.

As the reviewer's comment, we carefully corrected the typos and grammatical mistakes. The language of the manuscript was improved with the help of a native speaker. Corrected words and phrases are highlighted with yellow color.

3. Finally the applicability of the processing on flexible/r-2-r substrates is interesting but this is e.g. Fig 1 somewhat ambitious given the nature of the dew;/processing time and the feed rate typical with such systems. The reader should be able to infer this application without it being specified.

(A) To confirm the applicability of the method to roll-to-roll process, we designed a virtual roll-to-roll (R2R) system (Fig. A6). The plasma process time is critical factor for application to R2R process. In Fig. 2, the plasma time of 45 s is enough to get the NRs. Because the deposition rate of Ag layer did not have a significant effect on morphology or growth rate of AgCl NRs (Fig. A7), the deposition rate is set to be 20 Å/s. When the plasma chamber was 90 cm long and the Ag deposition chamber was 300 cm long, the feed rate of R2R process was 1.2 m/min, which is compatible with a commercial process.

$$\text{Size of plasma chamber} = (\text{Feed rate}) \times (\text{Plasma time}) = (1.2 \text{ m/min}) \times (45 \text{ s}) = 90 \text{ cm}$$

To deposit a 300-nm-thick Ag layer at a rate of 20 Å/s, the size of Ag deposition chamber should be 300 cm as in below.

$$\text{Size of deposition chamber} = \frac{(\text{Thickness of Ag})}{(\text{Deposition rate})} \times (\text{Feed rate}) = \frac{(3000 \text{ \AA})}{(20 \text{ \AA/s})} \times (1.2 \text{ m/min})$$

$$= 300 \text{ cm}$$

Those feed rate and chamber size are compatible with common roll-to-roll systems, and one can obtain AgCl NRs on polymer film with 2 cm/s.

Figure A6. Schematic illustration of roll-to-roll process to produce AgCl nanorods on plastic film.

Figure A7. SEM images of AgCl nanorods as a function of Ag deposition rate. Deposition rate did not have a significant effect on the morphology of nanorods.

We revised a sentence in the revised manuscript (page 13, line 24)

The plasma-induced AgCl NRs could be easily fabricated using a roll-to-roll process over a large area on any polymer film because the method does not require lithography, mask molds, or thermal processes.

→ To confirm the applicability of the method to R2R process, we designed a virtual roll-to-roll (R2R) system (Supplementary Fig. S10). The plasma process time is a critical factor for application to R2R process. The plasma time of 45 s is sufficient to get the NRs (Fig. 2). Because the deposition rate of Ag layer did not have a significant effect on morphology or growth rate of AgCl NRs (Supplementary Fig. S11), the deposition rate is set to be 20 Å/s. When the plasma chamber was 90 cm long, and the Ag-deposition chamber was 300 cm long, the feed rate of R2R process was 1.2 m/min, which could be used in a commercial process. As a result, the plasma-induced AgCl NRs could be rapidly fabricated using a R2R process over a large area on several kinds of polymer film because the method does not require lithography, mask molds, or thermal processes.

We added a supplementary figure S10 and S11

Supplementary Figure S10. Design of roll-to-roll system to fabricate AgCl nanorods. Schematic illustration of roll-to-roll process to produce AgCl nanorods on plastic film.

Supplementary Figure S11. Effect of deposition rate of Ag on AgCl NRs. SEM images of AgCl nanorods as a function of Ag deposition rate. Deposition rate did not have significant effect on morphology of nanorods.

We added a paragraph in the revised manuscript (Supplementary Figure S10)

→ To confirm the applicability of the method to R2R process, we designed a virtual R2R system (Supplementary Fig. S10). The plasma process time is critical factor for application to R2R process. In Fig. 2, the plasma time of 45 s is enough to get the NRs. Because the deposition rate of Ag layer did not have a significant effect on morphology or growth rate of AgCl NRs (Supplementary Fig. S11), the deposition rate is set to be 20 Å/s. When the plasma chamber was 90 cm long and the Ag deposition chamber was 300 cm long, the feed rate of R2R process was 1.2 m/min, which is compatible with a commercial process.

$$\text{Size of plasma chamber} = (\text{Feed rate}) \times (\text{Plasma time}) = (1.2 \text{ m/min}) \times (45 \text{ s}) = 90 \text{ cm}$$

To deposit a 300-nm-thick Ag layer at a rate of 20 Å/s, the size of Ag deposition chamber should be 300 cm:

$$\begin{aligned} \text{Size of deposition chamber} &= \frac{(\text{Thickness of Ag})}{(\text{Deposition rate})} \times (\text{Feed rate}) = \frac{3000 \text{ \AA}}{20 \text{ \AA/s}} \times (1.2 \text{ m/min}) \\ &= 300 \text{ cm} \end{aligned}$$

Those feed rate and chamber size are compatible with common R2R systems, and can produce AgCl NRs on polymer film at 2 cm/s.

Specific comments

4. Figures all - please remove the system or secondary scale bars on images, this is distracting and messy

(A) As reviewer's comment, we removed secondary scale bars on images.

The Figure 2 is revised in the manuscript

Figure 2. AgCl nanorods on different substrates. Time evolution of Ag film during Cl₂ plasma treatment on different thick substrates, (a) 100-μm-thick polyimide (PI), (b) 130-μm-thick polyethylene terephthalate (PET), and (c) 700-μm-thick soda lime glass. Scale bar: 1 μm.

→

Figure 2. AgCl nanorods on different substrates. Time evolution of Ag film during Cl₂ plasma treatment on different thick substrates, (a) 100-μm-thick polyimide (PI), (b) 130-μm-thick polyethylene terephthalate (PET), and (c) 700-μm-thick soda lime glass. Scale bar: 1 μm.

The Figure 3 is revised in the manuscript

Figure 3. Surface morphology of AgCl nanorods and its optical properties. (a) Scanning electron microscopy (SEM, 45° tilt view) images of AgCl nanostructures with sizes of 10 nm, 100 nm, 300 nm, and 500 nm Ag (from left to right). Plasma time was fixed to be 45 s. (b) Average periods and (c) average diameters of the AgCl nanostructures as a function of Ag thickness. (d) Total transmittance and (e) haze transmittance of bare PI film and various AgCl nanostructures.

→

Figure 3. Surface morphology of AgCl nanorods and its optical properties. (a) Scanning electron microscopy (SEM, 45° tilt view) images of AgCl nanostructures with sizes of 10 nm, 100 nm, 300 nm, and 500 nm Ag (from left to right). Plasma treatment was fixed at 45 s. (b) Average periods and (c) average diameters of AgCl nanostructures as a function of Ag thickness. (d) Total transmittance and (e) haze transmittance of bare PI film and various AgCl nanostructures.

Figure S2 is revised in the manuscript

Figure S2. Optimization of the plasma power for fabricating AgCl nanorods. The 300-nm-thick Ag coated on PI films were exposed to Cl_2 plasma with different plasma power for 45 s at 10 mTorr.

→

Figure S2. Optimization of the plasma power for fabricating AgCl nanorods. 300-nm-thick Ag coated on PI films were exposed to Cl_2 plasma with different plasma power for 45 s at 10 mTorr.

Figure S3 is revised in the manuscript

Figure S3. Optimization of the plasma process pressure for fabricating AgCl nanorods. The 300-nm-thick Ag coated on PI films were exposure to Cl₂ plasma with different process pressures for 45 s at plasma power of 350 W.

→

Figure S3. Optimization of the plasma process pressure for fabricating AgCl nanorods. 300-nm-thick Ag coated on PI films were exposed to Cl₂ plasma with different process pressures for 45 s at plasma power of 350 W.

5. Fig 6 and S6 - comment required in text about the emergence of the (111)(311)(222) peaks and how this fits the growth model proposed is required in the text.

(A) XRD phi scan was used to determine the crystal orientations of Ag and AgCl. To measure off-axis phi scans, we choose arbitrary reflection plane which is not surface normal plane. To allow interpretation of the in-plane relationship, c-Sapphire was used instead of PI film. The surface normal planes of c-sapphire, Ag, and AgCl are (006), (111), and (200) respectively. Thus, we performed phi scans along the c-sapphire (113) plane with psi angle (ψ) = 61.2°, Ag (311) with ψ = 29.5°, and AgCl (220) with ψ = 45°, where ψ is the angle between measured reflection plane of the phi scan and the surface normal plane of sample. In the phi scan, Ag film exhibited 6-fold symmetry on c-sapphire substrates with coincidence in angular positions (Fig. 5e). This result reveals that Ag film was deposited on c-sapphire with (111)-preferred orientation. After Cl₂ plasma treatment, Ag changed to AgCl. Ag maintains the 6-fold symmetry, but AgCl exhibits 12-fold symmetry with 30° intervals (Fig. 5f). Ag and AgCl had the same angular positions. AgCl (200) with 12-fold symmetry has three types of cubic domains, which are aligned on hexagonal Ag (111) planes with 60° azimuthal rotation (Fig. 5g). The phi scan confirmed that the growth direction of AgCl NRs was (200), which is in good agreement with HR-TEM observation (Fig. 5b)

The paragraph was revised in the revised manuscript (page 7, line 11 ~ page 7, line 21)

To further confirm the epitaxial relationship between Ag and AgCl, off-axis phi scans (azimuthal scan) were performed with θ fixed along the c-sapphire (113), Ag (311), and AgCl (220) reflections before and after Cl₂ plasma. To allow interpretation of the in-plane relationship, c-Sapphire was used instead of PI film. In the as-deposited Ag film (Fig. 5e), Ag film exhibited 6-fold symmetry on c-sapphire substrates with coincidence in angular positions. This result reveals that Ag film was deposited with (111) preferred orientation as shown in θ - 2 θ scans (Fig. 5d). After Cl₂ plasma in 45 s (Fig. 5f), Ag changed to AgCl, while maintaining 6-fold symmetry. Interestingly, AgCl exhibits 12-fold symmetry with 30° intervals, and the same angular position as the remaining Ag film. The 12-fold symmetry can be interpreted as cube-on-hexagon epitaxy (Fig. 5g). Three types of cube domains of AgCl (200) were aligned on hexagonal Ag (111) planes with 60° azimuthal rotation.

→ To further confirm the epitaxial relationship between Ag and AgCl, off-axis phi scans (azimuthal scan) were performed. XRD phi scan was used to determine the crystal orientations of Ag and AgCl. To measure off-axis phi scans, we choose arbitrary reflection plane which is not surface normal plane. To allow interpretation of the in-plane relationship, c-Sapphire was used instead of PI film. The surface normal planes of c-sapphire, Ag, and AgCl are (006), (111), and (200) respectively. Thus, we performed phi scans along the c-sapphire (113) plane with psi angle (ψ) = 61.2°, Ag (311) with ψ = 29.5°, and

AgCl (220) with $\psi = 45^\circ$, where ψ is the angle between measured reflection plane of the phi scan and the surface normal plane of sample. In the phi scan, Ag film exhibited 6-fold symmetry on c-sapphire substrates with coincidence in angular positions (Fig. 5e). This result reveals that Ag film was deposited on c-sapphire with (111)-preferred orientation. After Cl_2 plasma treatment, Ag changed to AgCl. Ag maintains the 6-fold symmetry, but AgCl exhibits 12-fold symmetry with 30° intervals (Fig. 5f). Ag and AgCl had the same angular positions. AgCl (200) with 12-fold symmetry has three types of cubic domains, which are aligned on hexagonal Ag (111) planes with 60° azimuthal rotation (Fig. 5g). The phi scan confirmed that the growth direction of AgCl NRs was (200), which is in good agreement with HR-TEM observation (Fig. 5b)

6. Fig S4 - overall quality of image a is very poor

(A) As reviewer's comment, we changed it to a high-quality image.

Figure S4 is revised in the manuscript

Supplementary Figure S4. Time evolution of Ag film during exposure to Cl_2 plasma. (a) SEM images, (b) Total optical transmittance and (c) Haze of of AgCl nanostructure as functions of Ag

→

Supplementary Figure S4. Time evolution of Ag film during exposure to Cl₂ plasma. (a) SEM images, (b) total optical transmittance and (c) haze of AgCl nanostructure as functions of Ag thickness and Cl₂ plasma exposure time.

7. Fig 1 - see proviso comments, I'm not sure a r-2-r system is good use of a figure and have to question the immediate suitability of the growth method and this technique.

(A) We answered that the growth method is suitable to industry R2R process in question #3.

To confirm the applicability of the method to roll-to-roll process, we designed a virtual roll-to-roll (R2R) system (Fig. A6). The plasma process time is critical factor for application to R2R process. In Fig. 2, the plasma time of 45 s is enough to get the NRs. Because the deposition rate of Ag layer did not have a significant effect on morphology or growth rate of AgCl NRs (Fig. A7), the deposition rate is set to be 20 Å/s. When the plasma chamber was 90 cm long and the Ag deposition chamber was 300 cm long, the feed rate of R2R process was 1.2 m/min, which is compatible with a commercial process.

$$\text{Size of plasma chamber} = (\text{Feed rate}) \times (\text{Plasma time}) = (1.2 \text{ m/min}) \times (45 \text{ s}) = 90 \text{ cm}$$

To deposit a 300-nm-thick Ag layer at a rate of 20 Å/s, the size of Ag deposition chamber should be 300 cm as in below.

$$\begin{aligned} \text{Size of deposition chamber} &= \frac{(\text{Thickness of Ag})}{(\text{Deposition rate})} \times (\text{Feed rate}) = \frac{(3000 \text{ \AA})}{(20 \text{ \AA/s})} \times (1.2 \text{ m/min}) \\ &= 300 \text{ cm} \end{aligned}$$

Those feed rate and chamber size are compatible with common roll-to-roll systems, and one can obtain AgCl NRs on polymer film with 2 cm/s.

Figure A6. Schematic illustration of roll-to-roll process to produce AgCl nanorods on plastic film.

Figure A7. SEM images of AgCl nanorods as a function of Ag deposition rate. Deposition rate did not have a significant effect on the morphology of nanorods.

We revised a sentence in the revised manuscript (page 13, line 24)

The plasma-induced AgCl NRs could be easily fabricated using a roll-to-roll process over a large area on any polymer film because the method does not require lithography, mask molds, or thermal processes.

→ To confirm the applicability of the method to R2R process, we designed a virtual roll-to-roll (R2R) system (Supplementary Fig. S10). The plasma process time is a critical factor for application to R2R process. The plasma time of 45 s is sufficient to get the NRs (Fig. 2). Because the deposition rate of Ag layer did not have a significant effect on morphology or growth rate of AgCl NRs (Supplementary Fig.

S11), the deposition rate is set to be 20 Å/s. When the plasma chamber was 90 cm long, and the Ag-deposition chamber was 300 cm long, the feed rate of R2R process was 1.2 m/min, which could be used in a commercial process. As a result, the plasma-induced AgCl NRs could be rapidly fabricated using a R2R process over a large area on several kinds of polymer film because the method does not require lithography, mask molds, or thermal processes.

We added a supplementary figure S10 and S11

Supplementary Figure S10. Design of roll-to-roll system to fabricate AgCl nanorods. Schematic illustration of roll-to-roll process to produce AgCl nanorods on plastic film.

Supplementary Figure S11. Effect of deposition rate of Ag on AgCl NRs. SEM images of AgCl

nanorods as a function of Ag deposition rate. Deposition rate did not have significant effect on morphology of nanorods.

We added a paragraph in the revised manuscript (Supplementary Figure S10)

→ To confirm the applicability of the method to R2R process, we designed a virtual R2R system (Supplementary Fig. S10). The plasma process time is critical factor for application to R2R process. In Fig. 2, the plasma time of 45 s is enough to get the NRs. Because the deposition rate of Ag layer did not have a significant effect on morphology or growth rate of AgCl NRs (Supplementary Fig. S11), the deposition rate is set to be 20 Å/s. When the plasma chamber was 90 cm long and the Ag deposition chamber was 300 cm long, the feed rate of R2R process was 1.2 m/min, which is compatible with a commercial process.

$$\text{Size of plasma chamber} = (\text{Feed rate}) \times (\text{Plasma time}) = (1.2 \text{ m/min}) \times (45 \text{ s}) = 90 \text{ cm}$$

To deposit a 300-nm-thick Ag layer at a rate of 20 Å/s, the size of Ag deposition chamber should be 300 cm:

$$\begin{aligned} \text{Size of deposition chamber} &= \frac{(\text{Thickness of Ag})}{(\text{Deposition rate})} \times (\text{Feed rate}) = \frac{3000 \text{ Å}}{20 \text{ Å/s}} \times (1.2 \text{ m/min}) \\ &= 300 \text{ cm} \end{aligned}$$

Those feed rate and chamber size are compatible with common R2R systems, and can produce AgCl NRs on polymer film at 2 cm/s.

8. Fig S1 shows and XRD pattern, not patterns. The line width of the plot hides key features of the diffraction peaks and should be reduced.

(A) As reviewer's comment, we reduced the line width of the XRD pattern in Figure S1 and its caption.

Figure S1 is revised in the manuscript

Figure S1. XRD patterns of the Cl₂ plasma exposed Ag. After Cl₂ plasma treatment on Ag, all of Ag was completely chlorized to produced AgCl compounds.

→

Figure S1. XRD pattern of Cl₂ plasma exposed Ag. After Cl₂ plasma treatment on Ag, all Ag had reacted completely to produce AgCl compounds.

9. Fig 3a - should be annotated to include the Ag thickness of the substrate.

(A) As reviewer's comment, Ag thicknesses were annotated in the each figure.

Figure 3 is revised in the manuscript

Figure 3. Surface morphology of AgCl nanorods and its optical properties. (a) Scanning electron microscopy (SEM, 45° tilt view) images of AgCl nanostructures with sizes of 10 nm, 100 nm, 300 nm, and 500 nm Ag (from left to right). The plasma time was fixed to be 45 s. (b) Average periods and (c) average diameters of the AgCl nanostructures as a function of Ag thickness. (d) Total transmittance and (e) haze (%) as a function of wavelength for different Ag thicknesses.

(e) haze transmittance of bare PI film and various AgCl nanostructures.

→

Figure 3. Surface morphology of AgCl nanorods and its optical properties. (a) Scanning electron microscopy (SEM, 45° tilt view) images of AgCl nanostructures with sizes of 10 nm, 100 nm, 300 nm, and 500 nm Ag (from left to right). Plasma treatment was fixed at 45 s. (b) Average periods and (c) average diameters of AgCl nanostructures as a function of Ag thickness. (d) Total transmittance and (e) haze transmittance of bare PI film and various AgCl nanostructures.

Reviewers' comments:

Reviewer #4 (Remarks to the Author):

The authors have answered the reviewers' questions and made corrections in the manuscript in a way which is acceptable for publication. Some errors still exist. For example on page 2 the sentence "Bottom-up synthesis of semiconductor NRs", is wrong. Bottom-up synthesis for semiconducting NRs can be done down to 60 degree C (or below), which the authors also mention in the tables in answering the reviewers. Too old references have been used by the authors. After the correction(s) I recommend publication of the manuscript.

Replies on reviewer's comments

We appreciate very much for your kind comments on our paper. We have responded point by point to the comments of the reviewers and revised the manuscript in accordance with the reviewers' comments. English was improved through a professional editorial service for grammatical correction. Corrected words and phrases are highlighted with red color through the manuscript to be recognized easily.

Replies on reviewer's comments are given below.

<For Reviewer #4>

The authors have answered the reviewers' questions and made corrections in the manuscript in a way which is acceptable for publication. Some errors still exist. For example on page 2 the sentence "Bottom-up synthesis of semiconductor NRs", is wrong. Bottom-up synthesis for semiconducting NRs can be done down to 60 degree C (or below), which the authors also mention in the tables in answering the reviewers. Too old references have been used by the authors. After the correction(s) I recommend publication of the manuscript.

(A) There are several kinds of bottom-up synthesis such as vapor-liquid-solid, chemical vapor deposition, thermal oxidation, and hydrothermal growth (Table A1). Most of them require high temperature ($> 500\text{ }^{\circ}\text{C}$), but few of them (ex. hydrothermal growth) don't require high temperature. Thus, we agree that the sentence "Bottom-up synthesis of semiconductor NRs requires growth temperatures $> 500\text{ }^{\circ}\text{C}$ " is wrong. As the reviewer's comment, we revised the incorrect sentence.

In the manuscript, too old references have been used, so we changed them to recent papers.

We revised a paragraph in the revised manuscript (page 2, line 10 ~ page 2, line 17)

Efforts to fabricate nanostructures on rigid substrates include top-down methods such as photo-,^{16,17} laser interference-,¹⁸ and e-beam lithography¹⁹; and bottom-up methods such as synthesis of semiconductor NRs by vapor-liquid-solid,²⁰⁻²² chemical vapor deposition,^{23,24} and strain relaxation²⁵. However, most of them are not suitable for fabricating nanostructure on flexible substrates. Lithography techniques use photoresists; their removal entails use of solvents that can damage a polymer film.^{26,27} Bottom-up synthesis of semiconductor NRs requires growth temperatures $> 500\text{ }^{\circ}\text{C}$,²⁰⁻²² but most plastic substrates cannot tolerate temperature $> 150\text{ }^{\circ}\text{C}$.²⁸

→ Efforts to fabricate nanostructures on rigid substrates include top-down methods such as photo-,^{16,17} laser interference-,¹⁸ and e-beam lithography¹⁹; and bottom-up methods such as synthesis of semiconductor NRs by vapor-liquid-solid (VLS),²⁰⁻²² chemical vapor deposition (CVD),^{23,24} and thermal oxidation. However, most of them are not suitable for fabricating nanostructure on flexible substrates. Lithography techniques use photoresists; their removal entails use of solvents that can damage a polymer film.^{26,27} VLS, CVD, and thermal oxidation require growth temperatures > 500 °C,²⁰⁻
²² but most plastic substrates cannot tolerate temperature > 150 °C.²⁸

As reviewer's comment, we changed too old references to recent references.

1. Forrest S. R. The path to ubiquitous and low-cost organic electronic appliances on plastic. *Nature* **428**, 911-918 (2004).

→ 1. White M. S., *et al.* Ultrathin, highly flexible and stretchable PLEDs. *Nat. Photonics* **7**, 811-816 (2013).

20. Wagner R., Ellis W. Vapor-liquid-solid mechanism of single crystal growth. *Appl. Phys. Lett.* **4**, 89-90 (1964).

→ 20. Klamchuen A., *et al.* Rational Concept for Designing Vapor-Liquid-Solid Growth of Single Crystalline Metal Oxide Nanowires. *Nano Lett.* **15**, 6406-6412 (2015).

21. Hu J., Odom T. W., Lieber C. M. Chemistry and physics in one dimension: synthesis and properties of nanowires and nanotubes. *Acc. Chem. Res.* **32**, 435-445 (1999).

→ 21. Cheng G., *et al.* Large anelasticity and associated energy dissipation in single-crystalline nanowires. *Nat. Nanotech.* **10**, 687-691 (2015).

22. Wu Y., *et al.* Inorganic semiconductor nanowires: rational growth, assembly, and novel properties. *Chem. Eur. J.* **8**, 1260-1268 (2002).

→ 22. Shen Y., *et al.* Epitaxy-Enabled Vapor-Liquid-Solid Growth of Tin-Doped Indium Oxide Nanowires with Controlled Orientations. *Nano Lett.* **14**, 4342-4351 (2014).

23. Bower C., Zhu W., Jin S., Zhou O. Plasma-induced alignment of carbon nanotubes. *Appl. Phys. Lett.* **77**, 830-832 (2000).
- 23. Annanouch F. E., *et al.* Aerosol-assisted CVD-grown WO₃ nanoneedles decorated with copper oxide nanoparticles for the selective and humidity-resilient detection of H₂S. *ACS Appl. Mater. Interfaces* **7**, 6842-6851 (2015).
24. Liu X., Wu X., Cao H., Chang R. Growth mechanism and properties of ZnO nanorods synthesized by plasma-enhanced chemical vapor deposition. *J. Appl. Phys.* **95**, 3141-3147 (2004).
- 24. Xu L., *et al.* Catalyst-free, selective growth of ZnO nanowires on SiO₂ by chemical vapor deposition for transfer-free fabrication of UV photodetectors. *ACS Appl. Mater. Interfaces* **7**, 20264-20271 (2015).
32. Tao A., *et al.* Langmuir-Blodgett silver nanowire monolayers for molecular sensing using surface-enhanced Raman spectroscopy. *Nano Lett.* **3**, 1229-1233 (2003).
- 35. Ariga K., Yamauchi Y., Mori T., Hill J. P. 25th Anniversary Article: What Can Be Done with the Langmuir-Blodgett Method: Recent Developments and its Critical Role in Materials Science. *Adv. Mater.* **25**, 6477-6512 (2013).
40. Weaver J. F., Hoflund G. B. Surface characterization study of the thermal decomposition of AgO. *J. Phys. Chem.* **98**, 8519-8524 (1994).
- 43. Mathpal M. C., *et al.* Opacity and plasmonic properties of Ag embedded glass based metamaterials. *RSC Adv.* **5**, 12555-12562 (2015).
44. Piao H., Adib K., Barteau M. A. A temperature-programmed X-ray photoelectron spectroscopy (TPXPS) study of chlorine adsorption and diffusion on Ag (111). *Surf. Sci.* **557**, 13-20 (2004).
- 47. Pan S.-D., *et al.* Controlled synthesis of pentachlorophenol-imprinted polymers on the surface of magnetic graphene oxide for highly selective adsorption. *J. Mater. Chem. A* **2**, 15345-15356 (2014).

47. Stoney G. G. The Tension of Metallic Films Deposited by Electrolysis. *Proceedings of the Royal Society of London. Series A, Containing Papers of a Mathematical and Physical Character* **82**, 172-175 (1909).
- 50. Janssen G., Abdalla M., Van Keulen F., Pujada B., Van Venrooy B. Celebrating the 100th anniversary of the Stoney equation for film stress: Developments from polycrystalline steel strips to single crystal silicon wafers. *Thin Solid Films* **517**, 1858-1867 (2009).
48. Saka M., Yamaya F., Tohmyoh H. Rapid and mass growth of stress-induced nanowhiskers on the surfaces of evaporated polycrystalline Cu films. *Scr. Mater.* **56**, 1031-1034 (2007).
- 25. Zhang Q., et al. Facile large-scale synthesis of vertically aligned CuO nanowires on nickel foam: Growth mechanism and remarkable electrochemical performance. *J. Mater. Chem. A* **2**, 3865-3874 (2014).